# Cross-Layer Discrete Concept Discovery for Interpreting Language Models

## Abstract

Interpreting language models remains challenging due to the existence of residual stream, which linearly mixes and duplicates features across adjacent layers, causing single-layer analyses to miss this cross-layer structure. Cross-layer sparse autoencoders (SAEs) address layer mixing but operate in continuous space, where concepts split across many neurons without clear boundaries. We introduce cross-layer vector quantized-variational autoencoder (CLVQ-VAE), a novel framework which maps representations from a lower layer to a higher layer through a discrete vector-quantization bottleneck, collapsing duplicated residual-stream features into compact, interpretable concept vectors. Our approach combines top-$k$ temperature-based sampling with exponential moving average (EMA) codebook updates, providing controlled exploration of the discrete latent space while maintaining codebook diversity. Across both encoder- and decoder-based models on ERASER-Movie, Jigsaw, and AGNews, CLVQ-VAE outperforms clustering, single-layer vector quantized-variational autoencoder (VQ-VAE), and sparse autoencoder (SAE) baselines across three evaluation axes: removing identified concepts drops model accuracy by up to 93%, LLM judges rank our concepts first in 66.7% of comparisons, and human annotators recover model predictions from our visualizations with 78% accuracy versus 54% for clustering. [1]

## 1 Introduction

Large language models (LLMs) have demonstrated remarkable capabilities across a wide range of natural language processing tasks, yet their internal mechanisms remain largely opaque. This opacity poses major challenges for interpretability, limiting scientific understanding and raising concerns around trust, accountability, and responsible use (Dodge et al., 2021; Sheng et al., 2021).

The majority of interpretability research focus on representations at individual layers — either by analysis of the activation patterns of neurons (Zhang et al., 2021) or by using probing classifiers that map the hidden states into pre-defined concepts (Belinkov et al., 2017; Arps et al., 2022; Kumar et al., 2023). These single-layer methods fail to account for how transformer residual streams duplicate and mix information across layers, obscuring computational structure that is only visible when multiple layers are examined together (Team, 2024).

Recent SAE methods (Härle et al., 2024; Lan et al., 2025) and transcoder architectures (Marks et al., 2024; Dunefsky et al., 2024a) have highlighted the value of analyzing layer pairs, with empirical studies showing cross-layer analysis yields more interpretable features than single-layer approaches (Shi et al., 2025; Balagansky et al., 2025; Laptev et al., 2025). This is largely due to the additive residual stream: each layer contributes to a running representation, causing features to persist and appear duplicated when layers are viewed in isolation (Lindsey et al., 2025). However, SAE-based methods operate in continuous spaces where concepts "split" across many neurons (Bricken et al., 2023), so a single neuron no longer corresponds to a discrete concept, forcing arbitrary thresholds or multi-vector combinations to isolate sparse activations (Oozeer et al., 2025; Bărbălau et al., 2025). Because humans reason in discrete categories, this misalignment limits interpretability (Wu et al., 2024).

---

[1]Code repository: `https://anonymous.4open.science/r/CLVQVAE-9386`

VQ-VAEs have been extensively explored in computer vision to discretize the continuous representations of images into codebook vectors (van den Oord et al., 2018; Takida et al., 2022; Razavi et al., 2019). We hypothesize that when VQ-VAEs are applied to language model activations, the codebook vectors will capture interpretable linguistic concepts — syntactic patterns or semantic categories — essential for the VQ-VAE reconstruction objective. Also, even though VQ-VAEs share the reconstruction objective of SAEs, they critically differ by utilizing a single discrete codebook vector rather than a linear combination of active neurons. This discrete bottleneck naturally concentrates information, sidestepping the ambiguity of identifying which feature directions to consider in combination.

Building on these insights, we propose CLVQ-VAE, a framework that discovers concepts across transformer layers. Unlike standard VQ-VAEs that reconstruct the same layer, our model acts as a transcoder, i.e., mapping activations from a lower layer $l$ to a higher layer $h$ through a discrete bottleneck, thus collapsing redundant residual-stream features into interpretable codebook vectors. We further improve this architecture by introducing a stochastic sampling mechanism that selects from the top-$k$ nearest codebook vectors using temperature-controlled probability distributions, resulting in better codebook utilization and concept diversity compared to deterministic approaches.

We evaluate CLVQ-VAE on the ERASER-Movie (Pang & Lee, 2004), Jigsaw Toxicity (cjadams et al., 2017), and AGNEWS (Gulli, 2005) datasets using fine-tuned RoBERTa (Liu et al., 2019b), BERT (Devlin et al., 2019), and decoder-only models like LLaMA-2-7b and Qwen2.5-3B-Instruct. Perturbation-based experiments shows that our approach identifies salient concepts that strongly influence the predictions, outperforming clustering, single-layer VQVAE, and SAE baselines. The quality of these concepts is further validated by an LLM-as-a-judge evaluation, which finds them more coherent than those from the competing methods. Finally, human evaluation confirms their practical utility for interpretation, with CLVQ-VAE visualizations achieving higher model-alignment and inter-annotator agreement scores than the clustering baseline.

## 2 Methodology

The CLVQ-VAE framework discovers concepts by reconstructing higher-layer activations from quantized lower-layer representations. As shown in Figure 1, it processes activations through three core components:

1. **Adaptive Residual Encoder:** This component applies controllable interpolation to input embeddings from a lower layer, preserving the semantic information in the pre-trained representations.

2. **Vector Quantizer:** This acts as a discrete bottleneck, mapping the continuous encoder outputs to one of the codebook vectors, forcing the model to represent information compactly.

3. **Transformer Decoder:** Lastly, this component takes the codebook vectors corresponding to the encoder output and reconstructs the target activations from a higher layer, hence learning to predict the model's cross-layer computations from the discrete concepts alone.

These components are jointly optimized through a reconstruction loss that encourages the encoder to map inputs to relevant codebook vectors and trains the decoder to reconstruct higher-layer activations from these quantized representations.

### 2.1 Problem Formulation

We formalize concepts as vectors in a learned codebook $\mathcal{E} = \{\mathbf{e}_j\}_{j=1}^{K}$, where $\mathbf{e}_j \in \mathbb{R}^d$ and each corresponds to a cluster of semantically related token representations. Unlike continuous sparse autoencoder activations, our approach enforces discrete assignments: each encoder output $\mathbf{z}_e$ from layer $l$ maps to its closest codebook vector $\mathbf{z}_q = \mathbf{e}_{j^*} \in \mathcal{E}$, where $j^* = \arg\min_j \|\mathbf{z}_e - \mathbf{e}_j\|_2$.

### 2.2 Adaptive Residual Encoder

Transformer-based language models produce rich, contextualized embeddings at each layer that have been shown to encode diverse linguistic information (Liu et al., 2019a; Sajjad et al., 2022b). A major challenge in

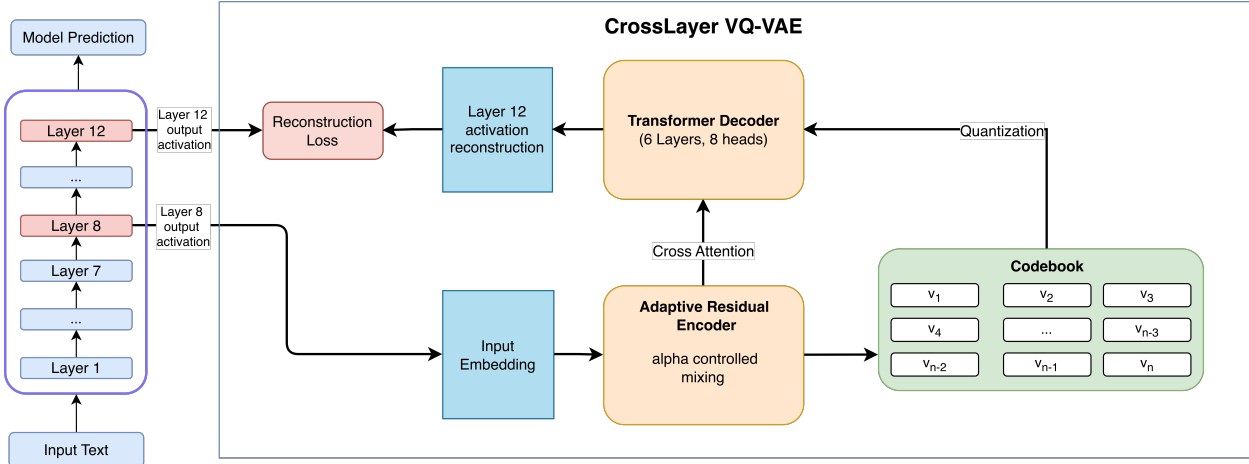

Figure 1: Overview of the CLVQ-VAE framework for cross-layer concept discovery. Lower-layer activations are passed through an adaptive residual encoder, discretized via vector quantization into concept vectors, and decoded to reconstruct higher-layer representations.

adapting the VQ-VAE encoder to language model embeddings lies in determining the appropriate amount of embedding changes. These changes must be sufficient to enable codebook reassignment for effective reconstruction, yet constrained enough to preserve the semantic knowledge encoded in the embeddings.

To address this, we propose an adaptive residual encoder that implements controlled manipulation of the input embeddings. Rather than completely transforming the input, which would destroy valuable linguistic features due to random initialization of encoder parameters, or leaving it unchanged, which would limit concept discovery, our encoder introduces a learnable interpolation mechanism that respects the information-rich nature of embeddings while enabling targeted refinements for cross-layer reconstruction.

Given an input embedding $\mathbf{x} \in \mathbb{R}^d$ from layer $l$, the encoder produces an output $\mathbf{z}_e$ through the following:

$$\mathbf{z}_e = (1 - \alpha) \cdot \mathbf{x} + \alpha \cdot \text{LN}(W\mathbf{x} + \mathbf{b}) \tag{1}$$

where $\alpha = \sigma(a) * 0.5$ is a mixing coefficient constrained to $[0, 0.5]$ with $a$ being a learnable parameter, and LN is the layer normalization applied to the linearly transformed $x$. We limit $\alpha$ to a maximum of 0.5 to prevent excessive modification of the original embedding, as we empirically found in Table 19 that allowing complete transformation ($\alpha \in [0,1]$) resulted in reduced codebook utilization.

In Table 19, we also observe that for an adaptive $\alpha$, the model initially prefers a small value of $\alpha$, which constrains the modification of the original embedding. As training progresses and encoder parameters get trained, the gradients gradually increase $\alpha$, allowing the model to make progressively larger modifications.

## 2.3 Vector Quantizer

The vector quantizer maps the encoder outputs to discrete concept representations. To promote the stable training and effective codebook utilization, we utilize three key mechanisms, i.e., k-means-based codebook initialization, temperature-controlled top-k sampling and the EMA-based codebook updates.

### 2.3.1 Codebook Initialization

The initial state of the codebook impacts training stability and concept quality. We explore two approaches: **random initialization** and **k-means initialization**, the latter with two variants. Each trades off simplicity, computational cost, and alignment with the data's underlying structure.

**Random Initialization:** A naive baseline that samples $K$ unique embedding vectors from the training dataset as initial codebook entries. While computationally inexpensive, the resulting codebook may not be representative of the data distribution, leading to slower convergence or "dead" codes.

**K-Means Initialization:** Initializes the codebook with cluster centroids of the input data, providing better coverage and typically leading to faster, more stable training. We evaluate two variants:

1. **Default K-Means.** Widely adopted in the VQ-VAE literature Łańcucki et al. (2020); Huh et al. (2023); Zeghidour et al. (2021), this variant applies standard k-means to all training embeddings from layer $l$. The resulting $K$ Euclidean centroids serve as the initial codebook vectors $\mathbf{e}_j$.

2. **Spherical K-Means.** This variant clusters vectors by angular similarity (cosine distance) rather than Euclidean distance, motivated by the observation that semantic similarity in NLP is often captured by vector direction (Banerjee et al., 2005). Embeddings $\mathbf{x}_i$ are first unit-normalized ($\hat{\mathbf{x}}_i = \mathbf{x}_i / \|\mathbf{x}_i\|_2$), then clustered on the hypersphere. The resulting unit-vector centroids $\mathbf{c}_j$ are rescaled by the average magnitude of their assigned vectors to reintroduce magnitude information:

$$\mathbf{e}_j = \mathbf{c}_j \cdot \frac{1}{|C_j|} \sum_{i \in C_j} \|\mathbf{x}_i\|_2 \tag{2}$$

where $C_j$ is the set of original vectors assigned to cluster $j$. This produces codebook vectors that group semantically similar words while preserving magnitude.

### 2.3.2 Top-k Temperature-Based Codebook Sampling

Our vector quantization mechanism employs temperature-based sampling (Takida et al., 2022) from the top-$k$ nearest codebook vectors. For each encoder output $\mathbf{z}_e$, we compute distances to all codebook vectors $\{\mathbf{e}_j\}_{j=1}^K$ as:

$$d(\mathbf{z}_e, \mathbf{e}_j) = \|\mathbf{z}_e - \mathbf{e}_j\|_2^2 \tag{3}$$

Rather than deterministically selecting the closest vector, we identify the top-$k$ nearest codebook vectors and sample from them using a temperature-controlled distribution:

$$p(j|\mathbf{z}_e) = \frac{\exp(-d(\mathbf{z}_e, \mathbf{e}_j)/\tau)}{\sum_{j' \in \text{top-}k} \exp(-d(\mathbf{z}_e, \mathbf{e}_{j'})/\tau)} \tag{4}$$

where $\tau$ is the temperature parameter. In our optimal configuration, we set $k = 5$ and $\tau = 1.0$, balancing exploration with exploitation. This controlled stochasticity during training encourages more uniform codebook utilization, reduces codebook collapse, and improves concept diversity (Appendix E.4).

### 2.3.3 EMA-Based Codebook Updates

To ensure the stable training dynamics, codebook is updated using the EMA (Łukasz Kaiser et al., 2018) rather than the direct backpropagation. The gradient-based updates can oscillate or become unstable with the discrete assignments (van den Oord et al., 2018), and thus we instead maintain the running averages of both the assignment counts and the accumulated vectors for each codebook entry.

For each codebook vector $\mathbf{e}_j$ in a training batch, we update its accumulated vector sum $\mathbf{m}_j$ and its total assignment count $N_j$ with a decay factor $\gamma = 0.99$. The codebook vector is then updated to the mean of its assigned embeddings by normalizing the accumulated sum by the count. This update strategy, when combined with our stochastic top-$k$ sampling, ensures the codebook remains diverse and active throughout the training while converging toward the stable representations of the cross-layer transformations. The complete update equations are provided in Appendix B.1.

## 2.4 Transformer Decoder

The final component of our CLVQ-VAE architecture is a transformer-based decoder that maps the sequence of the quantized representations to the higher-layer activations. It consists of 6 layers with 8 attention heads each and uses both the self-attention and the cross-attention mechanisms.

Because the decoder reconstructs the target activations for the entire input sequence at once rather than generating them sequentially, a causal mask is unnecessary. The self-attention is therefore fully bidirectional, allowing each token to draw the context from the entire sequence to build a more accurate representation.

The cross-attention mechanism allows the decoder to leverage information from the unquantized encoder outputs. Similar to the residual connection in skip-transcoders (Dunefsky et al., 2024b), which improves reconstruction without compromising interpretability, the cross-attention mechanism offloads low-level reconstruction details, enabling the codebook to focus on distinct, high-level concepts as demonstrated in auxiliary bottleneck models (Sheth & Kahou, 2023). We provide empirical evidence in Appendix E.1 that this cross-attention mechanism indeed improves interpretability and reconstruction.

## 2.5 Training Objectives

Our training incorporates two weighted objectives combined into a single loss function. The primary objective is the reconstruction loss, which minimizes the mean squared error between decoder output $\hat{\mathbf{y}}$ and target higher-layer representation $\mathbf{y}$, defined as $\mathcal{L}_{\text{rec}} = \|\mathbf{y} - \hat{\mathbf{y}}\|_2^2$ (Dunefsky et al., 2024a). This ensures that the model captures the transformations occurring between neural network layers. Additionally, we employ a commitment loss that encourages encoder outputs to commit to codebook vectors, calculated as $\mathcal{L}_{\text{commit}} = \|\mathbf{z}_e - \text{sg}(\mathbf{z}_q)\|_2^2$, where sg denotes the stop-gradient operator. The total loss is given by $\mathcal{L}_{\text{total}} = \mathcal{L}_{\text{rec}} + \beta\mathcal{L}_{\text{commit}}$, where $\beta$ controls the relative importance of the commitment term. We set $\beta = 0.1$ (van den Oord et al., 2018) to avoid constraining the encoder output too strictly (Wu & Flierl, 2019).

# 3 Experimental Setup

**Data.** We conduct experiments on three datasets: ERASER-Movie review dataset (Pang & Lee, 2004) for sentiment classification, Jigsaw Toxicity dataset (cjadams et al., 2017) for toxicity classification, and AGNEWS dataset (Gulli, 2005) for multi-class news categorization. The Appendix A.1 provides detailed dataset information.

**Model.** We use RoBERTa-base (Liu et al., 2019b) and BERT-base (Devlin et al., 2019) after fine-tuning on the respective datasets, and decoder-only models like LLaMA-2-7B (Touvron et al., 2023) and Qwen2.5-3B-Instruct with a task specific prompt and without finetuning. From these models, we extract paired activations: representations from a lower layer $l$ serve as input to CLVQ-VAE, while representations from a higher layer $h$ serve as reconstruction targets.

**Baseline.** For comparison, we include the clustering-based method from Yu et al. (2024), a single-layer VQ-VAE variant on layer $l$ (referred to as "Single-Layer"), and cross-layer sparse autoencoder (SAE) baselines. The single-layer SAE is included only in the faithfulness evaluation to serve as the SAE counterpart of Single-Layer, isolating the effect of the cross-layer objective for the SAE; the cross-layer SAE remains our primary SAE baseline for the other evaluations. Implementation details for each are provided in Appendix A.3.

**Activation Extraction.** We utilized the NeuroX toolkit (Dalvi et al., 2023) to extract paired activations from the four evaluated models. A key advantage of NeuroX is its ability to aggregate sub-word token representations of a model tokenizer into word-level activations. This aggregation enables the mapping of discovered concepts to complete words rather than fragmented tokens, allowing us to employ visualization techniques such as word clouds to represent and analyze the discrete concepts identified by CLVQ-VAE.

**Layer-Pair of Analysis.** For our primary evaluation across models and datasets, we apply CLVQ-VAE between intermediate-to-upper layer pairs: layers 8–12 for BERT/RoBERTa, layers 28–32 for LLaMA-2-

7B, and layers 32–36 for Qwen2.5-3B-Instruct. This specific layer pair is chosen based on theoretical and empirical evidence, which we detail in Appendix E.5.

# 4 Evaluation

We evaluate the concepts discovered by CLVQ-VAE through quantitative analysis (4.1), qualitative analysis (4.2), and an architectural design analysis (4.3). Each investigation is further supported by comprehensive supplementary material in Appendix C, Appendix D, and Appendix E, respectively.

## 4.1 Faithfulness Evaluation via Concept Ablation

We adopt the evaluation framework from Yu et al. (2024), which measures the faithfulness of discovered concepts by ablating their representations from sentence embeddings and measuring the impact on model performance for the task.

### 4.1.1 Methodology

We evaluate the efficacy of CLVQ-VAE in identifying salient concepts through concept ablation experiments. For each sentence, we identify the most salient token, find the codebook vector it maps to, and remove that vector from the sentence representation at layer $l$. The resulting drop in probe performance measures concept faithfulness.

To identify the most salient token in each input sentence, we use Layer Integrated Gradients (Layer IG) (Sundararajan et al., 2017b). This attribution method quantifies the contribution of each token's embedding to the model's final prediction, and the token with the highest attribution score is considered the most salient. Our model-specific implementations are detailed in the Appendix B.2.

We then construct three variants of the sentence representation embeddings:

- **Original CLS**: Unmodified sentence representation.

- **Perturbed CLS**: Sentence representation with the most salient concept removed via orthogonal projection (see Appendix B.3). For VQ-VAE-based, the concept vector is the nearest codebook vector. For clustering, it is the the closest cluster centroid, and for SAE, it is the subspace spanned by the encoder weight vectors corresponding to the top-$k$ activated neurons.

- **Random perturbed CLS**: Sentence representation with a random direction removed via orthogonal projection. This serves as a sanity check that performance drops are due to removing meaningful concepts rather than arbitrary perturbations.

To measure perturbation impact, we train a linear probe on the original CLS representations/task-labels via stratified cross-validation, and then evaluate each fold's held-out sentences for all three representation variants. If the identified concept is faithful, removing it (Perturbed CLS) should degrade accuracy more than removing a random direction (Random Perturbed CLS); see Appendix C.1 for implementation details.

As an additional control, we also evaluate against removing a randomly sampled active codebook direction — an active codebook vector assigned to other tokens but not the current salient token — to verify that drops are specific to the identified concept rather than any learned direction (Appendix C.4).

For encoder-based models like BERT and RoBERTa, we use the classification token ([CLS]) embedding as the sentence representation. For decoder-only models like LLaMA and Qwen, which lack a classification token, we use the mean of token embeddings (Lin et al., 2025). Throughout the paper, we will refer sentence representations as "CLS" for notational convenience, regardless of the underlying architecture.

### 4.1.2 Results: Baseline Comparison

Table 1 shows that CLVQ-VAE achieves the lowest perturbed accuracy in 5 of 12 settings, and a VQ-VAE method (CLVQ-VAE or Single-Layer) ranks first or second in 11 of 12 configurations. Drops under Perturbed

Table 1: Faithfulness comparison of methods across models and datasets. Lower perturbed accuracy indicates more faithful concept identification. Values in parentheses show percentage accuracy drop after perturbation. **Best** and second-best results are highlighted.

| Dataset | Method | RoBERTa | BERT | Llama | Qwen |
|---|---|---|---|---|---|
| **ERASER-Movie** | Clustering | 0.6271 (28.6%) | 0.7757 (6.0%) | **0.7779 (14.1%)** | 0.6740 (22.8%) |
| | Single-Layer | 0.0633 ± 0.0022 (92.8%) | 0.7624 ± 0.0127 (7.6%) | 0.8028 ± 0.0169 (11.3%) | 0.6142 ± 0.0080 (29.6%) |
| | Single-Layer SAE | 0.2218 ± 0.0162 (74.7%) | 0.7371 ± 0.0081 (10.6%) | 0.9088 ± 0.0007 (0.3%) | 0.8634 ± 0.0099 (1.9%) |
| | Cross-Layer SAE | 0.4361 ± 0.1664 (50.3%) | **0.4657 ± 0.0733 (43.5%)** | 0.9092 ± 0.0007 (0.3%) | 0.8731 ± 0.0014 (0.7%) |
| | **CLVQ-VAE** | **0.0594 ± 0.0009 (93.2%)** | 0.5311 ± 0.0354 (35.6%) | 0.7851 ± 0.0606 (13.3%) | **0.6113 ± 0.0241 (30.0%)** |
| **Jigsaw** | Clustering | **0.5628 (38.3%)** | 0.7577 (15.8%) | 0.7793 (7.5%) | 0.6352 (23.6%) |
| | Single-Layer | 0.9019 ± 0.0036 (1.1%) | 0.8079 ± 0.0077 (10.2%) | **0.7622 ± 0.0199 (9.6%)** | 0.5917 ± 0.0230 (28.8%) |
| | Single-Layer SAE | 0.9185 ± 0.0022 (-0.7%) | 0.8999 ± 0.0027 (0.0%) | 0.8412 ± 0.0015 (0.3%) | 0.8459 ± 0.0058 (0.3%) |
| | Cross-Layer SAE | 0.9172 ± 0.0022 (-0.6%) | 0.8335 ± 0.0064 (7.3%) | 0.8410 ± 0.0012 (0.3%) | 0.8463 ± 0.0029 (0.2%) |
| | **CLVQ-VAE** | 0.6127 ± 0.0421 (32.8%) | **0.7372 ± 0.0090 (18.0%)** | 0.7931 ± 0.0101 (5.9%) | **0.5809 ± 0.0121 (30.1%)** |
| **AGNEWS** | Clustering | 0.3875 (46.7%) | 0.6675 (10.5%) | **0.8485 (4.7%)** | 0.7542 (15.0%) |
| | Single-Layer | 0.1011 ± 0.0021 (86.1%) | 0.6711 ± 0.0500 (10.0%) | 0.8744 ± 0.0142 (1.8%) | **0.7164 ± 0.0241 (19.3%)** |
| | Single-Layer SAE | 0.1325 ± 0.0128 (81.8%) | **0.2656 ± 0.0147 (64.4%)** | 0.8989 ± 0.0012 (0.0%) | 0.8944 ± 0.0017 (0.3%) |
| | Cross-Layer SAE | 0.3280 ± 0.0807 (54.9%) | 0.3344 ± 0.0103 (55.2%) | 0.8995 ± 0.0010 (-0.1%) | 0.8992 ± 0.0009 (-0.2%) |
| | **CLVQ-VAE** | **0.0992 ± 0.0035 (86.4%)** | 0.6492 ± 0.0442 (13.0%) | 0.8758 ± 0.0028 (1.6%) | 0.7536 ± 0.0040 (15.1%) |

Table 2: Codebook concept specificity (RoBERTa), with random-assignment baseline in parentheses. *SL/DL Ratio* denotes same- vs. different-label Jaccard overlap at TF-IDF cosine $\geq 0.1$.

| Dataset | Label Purity | Label-Div. Tok. | Mean JSD | SL/DL Ratio |
|---|---|---|---|---|
| ERASER-Movie | 0.691 (0.50) | 75.8% | 0.848 | 4.26× |
| Jigsaw | 0.646 (0.50) | 32.1% | 0.916 | 7.56× |
| AGNews | 0.353 (0.25) | 43.7% | 0.823 | 2.10× |

CLS consistently exceed those under the random active codebook control (Appendix C.4), confirming that the identified concept directions are task-relevant rather than an artifact of the projection operation.

Both SAE variants show inconsistent behavior: near-zero or negative drops for all decoder models, and mixed results for encoder models, which we attribute to concept splitting across features (Bricken et al., 2023). CLVQ-VAE outperforms the cross-layer SAE in 10 of 12 configurations, including all 6 decoder-model settings by an average of 15.8 percentage points. Similar failure on decoder models is observed with an alternative SAE implementation (LlamaScope (He et al., 2024)), suggesting this is a general limitation of sparse representations within concept ablation-based faithfulness evaluation (Appendix C.3)

**Implementation Note:** (1) All results are averaged across 3 random seeds, with reference baseline values (Original CLS, Random Perturbation) in Appendix C.2. (2) Clustering has no standard deviation since hierarchical clustering is deterministic. (3) CLVQ-VAE and Single-Layer use k-means initialization (see §4.3.1). (4) Both SAE variants use top-$k = 10$; a full ablation over $k \in \{1, 5, 10\}$ is in Appendix A.3.2.

## 4.2 Interpretability Evaluation

We evaluate the plausibility and interpretability of identified codebook vectors as concepts through three analyses: (1) a structural analysis showing they are concept-specific rather than sentence-level compressors, (2) an LLM-as-a-judge evaluation of semantic coherence, and (3) a human study of concept interpretability.

### 4.2.1 Codebook Concept Specificity

We investigate whether the learned codebook vectors encode distinct semantic concepts or merely compress sentence representations through three analyses at the vector, token, and sentence level. Pairwise cosine similarity between codebook vectors, a geometric measure of distinctness, is reported in Appendix E.1.

**Methodology.** We perform three analyses on the trained codebook. At the **vector level**, we examine the label distribution of tokens assigned to each vector. At the **token level**, we test whether the same surface token routes to different vectors depending on sentence label, indicating concept-level rather than surface-form encoding. At the **sentence level**, we compare codebook assignments of same-label vs. different-label sentence pairs above a lexical similarity threshold. See Appendix D.5 for details.

**Evaluation Metrics.** *Label purity* (vector-level) is the fraction of tokens assigned to a vector that belong to its dominant label class, with baseline $1/|\mathcal{Y}|$ under random assignment (0.50 for binary tasks, 0.25 for the four-class AGNews), where $\mathcal{Y}$ is the set of label classes. *Jensen-Shannon divergence* (Lin, 1991) (JSD, token-level) measures how much a token's codebook assignment varies across label classes; tokens with differing dominant vectors across label pairs are labeled *label-divergent*. *TF-IDF cosine similarity* (Sparck Jones, 1972) (sentence-level) measures lexical similarity between sentence pairs, with threshold $\tau$ controlling the strictness of the match, and the *same-label/different-label Jaccard ratio* (sentence-level) measures the ratio of median codebook overlap between same-label and different-label pairs. Full definitions are in Appendix D.5.

**Results.** Across all twelve model-dataset configurations (full results in Table 17; representative RoBERTa values in Table 2), all three analyses show that codebook vectors encode label-discriminative concepts rather than acting as sentence-level compressors.

At the vector level, label purity exceeds the random baseline expected under undifferentiated compression (0.60–0.76 vs. 0.50 for binary tasks; 0.35–0.47 vs. 0.25 for AGNews), with a notable fraction of vectors achieving purity 1.0, firing exclusively on tokens from a single class (Figure 2).

At the token level, 28–76% of content tokens show label-divergent routing, with mean JSD of 0.56–0.92. For instance, in ERASER-Movie, *entertainment* routes to vector #137 (91% negative-class purity) in a negative review but to vector #73 (97% positive-class purity) in a positive one (further examples in Table 16).

At the sentence level, same-label pairs share significantly more codebook vectors than different-label pairs of comparable vocabulary, with overlap ratios of 2.10–7.56×. The gap widens at stricter thresholds: for Jigsaw/RoBERTa at $\tau = 0.3$, same-label pairs share 30.56× more vectors than different-label pairs.

### 4.2.2 LLM-as-a-Judge Evaluation

While faithfulness quantifies functional importance, it does not reveal semantic coherence. We use LLM-based evaluation to assess whether discovered concepts plausibly explain model predictions.

**Methodology.** Given an input text, its prediction, and the concept representation for its most salient token, we use LLM judges to evaluate concept quality. Each judge receives: (1) the input text, (2) the model's predicted class with its semantic label (e.g., "Positive" for class 1 in sentiment analysis, "Sports" for a news category), and (3) concept representations from all methods being compared.

To reduce individual biases from a single LLM, we use an ensemble of four LLMs (GPT-4o-mini, Claude 3.5 Haiku, Gemini 2.0 Flash, and Gemini 2.0 Flash Lite). We use stratified sampling to ensure balanced representation across prediction categories: for binary tasks, we sample equally from true/false positives/negatives; for multi-class tasks, we stratify by correctness.

We construct concept representations as follows: for each test instance, we identify its most salient token via Layer Integrated Gradients (Section 4.1.1), find which concept (codebook vector, cluster centroid, or SAE neuron) this token was assigned to, and retrieve all training tokens assigned to that concept. If the concept represents more than half of the [CLS] token representation, we present it as up to 5 exemplar sentences (5–30 words each); otherwise, we extract the 10 most frequent words. Concepts with no assigned training tokens receive an empty representation and an automatic rating of 1.

**Evaluation Metrics.** We assess concept quality using four metrics: *mean rating* (average score across judges and instances), *mean reciprocal rank* (MRR, measuring consistent top performance), *win rate* (proportion of configurations where a method ranks first), and *Kendall's W* (inter-judge agreement, $W \geq 0.7$ indicates strong consensus (Kendall & Babington Smith, 1939)). Full definitions are in Appendix D.1.

Table 3: LLM-judge based evaluation of methods across models and datasets.

| Method | Mean Rating $\pm$ Std | MRR | Win Rate |
|---|---|---|---|
| CLVQ-VAE | **1.890 $\pm$ 0.877** | **0.611** | **66.7%** |
| Single-Layer | 1.823 $\pm$ 0.874 | 0.458 | 44.4% |
| Cross-Layer SAE | 1.800 $\pm$ 0.888 | 0.542 | 44.4% |
| Clustering | 1.675 $\pm$ 0.855 | 0.472 | 44.4% |

**Results.** Table 3 summarizes LLM-judge based evaluation results. CLVQ-VAE achieves the best performance with a mean rating of 1.890 $\pm$ 0.877, MRR of 0.611, and win rate of 66.7%, consistently ranking first or second. Single-Layer shows competitive mean performance (1.823 $\pm$ 0.874) but its lower MRR (0.458) and win rate (44.4%) indicate it less frequently produces top-ranked concepts. Both CLVQ-VAE and Single-Layer used spherical initialization for this baseline comparison. The remaining baselines show weaker overall performance: Cross-Layer SAE obtains a mean rating of 1.800 $\pm$ 0.888 and Clustering 1.675 $\pm$ 0.855, with both achieving a win rate of 44.4%.

We calculate the inter-judge agreement via Kendall's coefficient of concordance. The results (detailed in Appendix D.4) show moderate consensus ($W = 0.533$ overall), with strongest agreement for Jigsaw ($W = 0.900$). ERASER-Movie ($W = 0.475$) and AGNews ($W = 0.225$) show weaker agreement, though the overall ranking of methods remains consistent across judges.

### 4.2.3 Human Evaluation

To complement our LLM-based analysis, we conducted a human evaluation study comparing CLVQ-VAE with the clustering baseline. This evaluation assesses whether discovered concepts can be effectively visualized and interpreted by humans.

**Methodology.** We randomly select 19 sentences from the ERASER-Movie review dataset, with roughly equal representation of true positives (TP), true negatives (TN), false positives (FP), and false negatives (FN), to evaluate concepts underlying correct and incorrect predictions. The rationale for this sample size and stratified design is detailed in Appendix D.2. For each sentence, we generate word clouds for CLVQ-VAE and clustering using the tokens mapped to the codebook vector associated with the sentence's most salient token. Examples across all four prediction categories are in Appendix F.

14 annotators participated in the study, collectively reviewing 266 samples. Each annotator saw the word cloud for each sentence without the original sentence, model prediction, or ground truth label. This way, annotators relied solely on the visualizations to infer model behavior. For each visualization, annotators:

1. Predict the model's sentiment label by only using the information in the word cloud.

2. Rate their confidence in this prediction on a scale of 1-10 (10 is highest).

**Evaluation Metrics.** We assess the visualization quality using three metrics: *Fleiss' Kappa* (inter-annotator agreement beyond chance, ranging from -1 to 1), *average confidence* (mean annotator certainty on a 1-10 scale) and *model alignment rate* (percentage of annotator predictions matching the model's actual prediction, regardless of correctness). Full metric formulations are provided in Appendix D.1.2.

**Results.** Table 4 presents the results of the human evaluation comparing the clustering approach (Yu et al., 2024) with our CLVQ-VAE framework. CLVQ-VAE achieved substantially higher inter-annotator agreement ($\kappa = 0.864$, "almost perfect agreement") compared to clustering ($\kappa = 0.59$, "moderate agreement"), suggesting more consistent interpretations across annotators. Annotators also reported higher confidence (8.44 vs. 5.981) when interpreting CLVQ-VAE visualizations, indicating greater conceptual clarity. Additionally, model alignment was over 24 percentage points higher (78.20% vs. 54.14%), showing that CLVQ-VAE more

Table 4: Human evaluation results comparing CLVQ-VAE with baseline clustering approach for ERASER-Movie dataset. Higher values indicate better performance across all metrics.

| Method | Fleiss' Kappa ($\kappa$) | Avg. Confidence | Model Alignment Rate |
|---|---|---|---|
| Clustering | 0.59 | 5.981 | 54.14% |
| CLVQ-VAE | **0.864** | **8.44** | **78.20%** |

Table 5: Faithfulness comparison of different CLVQ-VAE initialization methods across models and dataset.

| Model | Dataset | Spherical | K-means | Random |
|---|---|---|---|---|
| BERT | ERASER-Movie | $0.6188 \pm 0.0207$ (25.0%) | $\mathbf{0.5311 \pm 0.0354}$ **(35.6%)** | $0.5869 \pm 0.0210$ (28.8%) |
| | Jigsaw | $0.7637 \pm 0.0631$ (15.1%) | $0.7372 \pm 0.0090$ (18.0%) | $\mathbf{0.7283 \pm 0.0123}$ (19.0%) |
| | AGNEWS | $\mathbf{0.6303 \pm 0.0566}$ (15.5%) | $0.6492 \pm 0.0442$ (13.0%) | $0.6644 \pm 0.0395$ (10.9%) |
| RoBERTa | ERASER-Movie | $0.0598 \pm 0.0006$ (93.2%) | $\mathbf{0.0594 \pm 0.0009}$ **(93.2%)** | $0.0603 \pm 0.0011$ (93.1%) |
| | Jigsaw | $0.7176 \pm 0.0240$ (21.3%) | $\mathbf{0.6127 \pm 0.0421}$ **(32.8%)** | $0.7171 \pm 0.1033$ (21.4%) |
| | AGNEWS | $0.1036 \pm 0.0045$ (85.8%) | $\mathbf{0.0992 \pm 0.0035}$ **(86.4%)** | $0.1032 \pm 0.0005$ (85.8%) |
| LLaMA-2-7b | ERASER-Movie | $0.8508 \pm 0.0326$ (6.0%) | $\mathbf{0.7851 \pm 0.0606}$ **(13.3%)** | $0.8608 \pm 0.0251$ (4.9%) |
| | Jigsaw | $\mathbf{0.7819 \pm 0.0165}$ **(7.2%)** | $0.7931 \pm 0.0101$ (5.9%) | $0.8066 \pm 0.0128$ (4.3%) |
| | AGNEWS | $0.8826 \pm 0.0094$ (0.8%) | $\mathbf{0.8758 \pm 0.0028}$ **(1.6%)** | $0.8942 \pm 0.0075$ (-0.5%) |
| Qwen2.5-3B | ERASER-Movie | $0.6273 \pm 0.0099$ (28.1%) | $\mathbf{0.6113 \pm 0.0241}$ **(30.0%)** | $0.6189 \pm 0.0295$ (29.1%) |
| | Jigsaw | $\mathbf{0.5606 \pm 0.0382}$ **(32.6%)** | $0.5809 \pm 0.0121$ (30.1%) | $0.6096 \pm 0.0341$ (26.7%) |
| | AGNEWS | $0.6883 \pm 0.0429$ (22.4%) | $0.7536 \pm 0.0040$ (15.1%) | $\mathbf{0.6525 \pm 0.0621}$ **(26.5%)** |

faithfully communicates the model's reasoning. This rate covers all 19 sentences, including the FP and FN cases. Annotators thus recover the model's prediction even when it is wrong, indicating that CLVQ-VAE reflects the model's internal logic—including its mistakes—rather than just surface sentiment.

### 4.3 Design Choice Analysis

### 4.3.1 Codebook Initialization

We evaluate three initialization strategies — random, k-means, and spherical k-means — across models and datasets to assess their impact on quantitative performance and qualitative concept coherence.

**Quantitative Analysis (Faithfulness).** Table 5 presents faithfulness results across initialization methods. Default k-means achieves lowest perturbed accuracy in 7 of 12 model-dataset combinations, often outperforming the spherical variant. This suggests Euclidean distance-based partitioning may be more effective than angular similarity for embedding space organization in this context. Both k-means variants generally outperform random initialization, which introduces additional variance from its unstructured codebook.

**Qualitative Analysis (Interpretability).** Table 6 summarizes LLM judge evaluation results across initialization methods. Spherical initialization performs best, winning 62.5% of configurations and achieving the highest mean rating with the lowest variance. K-means performs comparably in mean rating but with slightly higher variance, suggesting less consistent quality. Random initialization shows substantially weaker performance, particularly in MRR (0.472) and win rate (25.0%). We observe strong inter-judge agreement (detailed in Appendix D.4) with an overall $W = 0.793$, where 9 of 12 configurations achieve $W \geq 0.7$.

The contrast between quantitative faithfulness metrics and qualitative evaluation might be revealing a practical consideration here: while k-means identifies functionally important features affecting model decisions, spherical initialization produces concepts that better align with human interpretation. This suggests initialization choice depends on the primary goal — functional faithfulness or semantic coherence.

Table 6: LLM-judge based evaluation of initialization methods across models and datasets.

| Method | Mean Rating $\pm$ Std | MRR | Win Rate |
|---|---|---|---|
| Spherical | **1.903 $\pm$ 0.306** | **0.694** | **62.5%** |
| K-means | 1.841 $\pm$ 0.330 | 0.667 | 58.3% |
| Random | 1.800 $\pm$ 0.390 | 0.472 | 25.0% |

Table 7: Impact of codebook size (K) on perturbed CLS accuracy and codebook perplexity for ERASER-Movie on RoBERTa model

.

| Codebook Size (K) | Perturbed CLS | Perplexity (Utilization %) |
|---|---|---|
| 50 | 0.0769 | 30.121 (60.2%) |
| 100 | 0.0665 | 49.600 (49.6%) |
| 400 | 0.0606 | 139.208 (34.8%) |
| 800 | 0.1214 | 168.444 (21.1%) |
| 1200 | 0.1004 | 184.895 (15.4%) |

### 4.3.2 Codebook Size

Table 7 shows the impact of codebook size $K$ on performance and utilization for the ERASER-Movie–RoBERTa configuration. Although perturbed accuracy varies only slightly, perplexity reveals meaningful trends in codebook usage. At $K = 400$, CLVQ-VAE achieves a perplexity of 139.208, striking a good balance between capacity and efficiency. Smaller codebooks (e.g., $K = 50$) achieve high utilization but force multiple concepts to share the same vector, reducing interpretability. Larger codebooks ($K = 800$ or $K = 1200$) underutilize available entries and may fragment the representation space.

### 4.3.3 Other Design Choices

**Commitment Loss Weight.** Following van den Oord et al. (2018), we set commitment cost $\beta = 0.1$. Our ablations (Appendix E.3) confirm this balances encoder-codebook alignment with representation flexibility: higher values ($\beta \geq 0.6$) over-constrain assignments and reduce perplexity below 82, while lower values maintain diversity but risk training instability.

**Sampling Parameters.** For stochastic sampling, we use temperature $\tau = 1.0$ and top-$k = 5$. While validation perplexity varies from 207 to 220 across different temperature settings (Appendix E.4), perturbed accuracy remains stable (0.0783 to 0.0911), showing limited sensitivity to these parameters. We therefore adopt conservative values that prioritize stable training dynamics while maintaining codebook diversity.

## 5 Related Work

Traditional interpretability methods for NLP, such as gradient-based and perturbation-based techniques (Sundararajan et al., 2017a; Kapishnikov et al., 2021; Rajagopal et al., 2021; Zhao & Aletras, 2023), assess input feature contributions to predictions but often fail to reveal internal decision-making processes. Representation analysis instead examines whether predefined concepts are learned in the representation and how such knowledge is structured in the model's neurons (Dalvi et al., 2019; Sajjad et al., 2022a; Gurnee et al., 2023), though it requires predefined concepts and annotated probing data (Antverg & Belinkov, 2022). The polysemantic and superpositional nature of neurons further complicates the neuron-level interpretation (Haider et al., 2025; Elhage et al., 2022; Fan et al., 2023).

Concept-based approaches aim to address some of these limitations by interpreting the model behavior through high-level concepts that are human-understandable. Techniques like TCAV measure the model sensitivity to predefined concepts via directional derivatives in the activation space (Kim et al., 2018), though they still rely on the manually specified concepts. Recent methods move toward discovering the latent concepts directly from the internal representations, which enables a deeper and more flexible understanding of the model functionality (Ghorbani et al., 2019; Dalvi et al., 2022; Jourdan et al., 2023; Yu et al., 2024).

Researchers have utilized sparse autoencoders (SAEs) (Härle et al., 2024) to extract interpretable features from the large language models (LLMs). However, studies have highlighted the challenges in their stability and utility. For instance, SAEs trained with different random seeds on the same data can learn divergent feature sets, which indicates the sensitivity to initialization (Paulo & Belrose, 2025). Furthermore, their performance on the downstream tasks does not consistently surpass the baseline methods, which questions their practical benefits (Kantamneni et al., 2025).

Cross-layer interpretability has gained attention, with researchers introducing sparse crosscoders to capture features across model layers (Lindsey et al., 2024). These methods facilitate tasks like model diffing and circuit analysis by tracking shared and unique features across layers, providing deeper insights into model behavior (Minder et al., 2025; Dunefsky et al., 2024b).

While VQ-VAE have shown success in domains like image and speech processing (van den Oord et al., 2018; Łukasz Kaiser et al., 2018; Huang et al., 2023; Guo et al., 2020; Huang & Ji, 2020; Bhardwaj et al., 2022), their application to NLP interpretability remains underexplored. The CLVQ-VAE framework addresses this gap by integrating transcoder-inspired objectives to map lower-layer representations to higher-layer ones.

## 6 Conclusion

We present CLVQ-VAE, a framework for discovering discrete concepts across transformer layers via vector quantization, collapsing redundant residual-stream features into interpretable codebook vectors. Across four language models and three datasets, CLVQ-VAE outperforms clustering, single-layer VQ-VAE, and SAE baselines in identifying functionally important and semantically coherent concepts.

Removing CLVQ-VAE-identified concepts drops model accuracy by up to 93.2%. LLM judges rank our concepts first in 66.7% of comparisons, and human annotators recover model predictions from our visualizations 78% of the time versus 54% for clustering, with higher inter-annotator agreement. These results validate that CLVQ-VAE discovers concepts that both influence predictions and align with human understanding.

Our design choices prove essential: the adaptive residual encoder balances knowledge preservation with refinement, while the cross-attention mechanism ensures the capture of distinct, task-critical concepts. Furthermore, temperature-based top-$k$ sampling maintains codebook diversity. We also uncover a trade-off in initialization, where k-means favors functional faithfulness and spherical k-means enhances semantic coherence. Overall, by integrating discrete representation learning with cross-layer analysis, CLVQ-VAE provides a robust framework for translating opaque model mechanisms into faithful, interpretable concepts.

## 7 Limitations

**Resource Demands.** Extracting activation pairs from multiple layers requires substantial memory, especially for larger models. K-means initialization adds further computational costs that scale with dataset size and codebook dimensions. Models exceeding 70B parameters may need further optimization.

**Evaluation Precision.** Our perturbation-based faithfulness measurement differentiates methods but shows limited sensitivity to hyperparameter changes like temperature, top-$k$, and codebook dimensions (see Appendix E.4 for details). Also, the LLM-based evaluation is sensitive to prompt construction and may struggle with unusual cases, requiring careful prompt refinement for consistency across different inputs.

**Architectural Transferability.** The framework requires training separate models for each layer combination, with optimal pairings varying by architecture. For decoder-only models, we use mean-pooled embeddings in place of CLS tokens, though they may encode different information than classification tokens.

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

## Appendix Contents

## A Experimental Setup

### A.1 Dataset

Table 8: The data size of each benchmark used in the evaluation: the ERASER Sentiment dataset, Jigsaw Toxicity dataset, and the AGNEWS dataset.

| Benchmark | Train | Dev | Tags |
|-----------|-------|------|------|
| ERASER    | 13878 | 856  | 2    |
| JIGSAW    | 9000  | 800  | 2    |
| AGNEWS    | 16000 | 1200 | 4    |

### A.2 Hyperparameters

Table 9 lists all hyperparameters used in our experiments. All weights use standard PyTorch random initialization.

Table 9: Hyperparameters used across all experiments.

| Category | Component | Value |
|----------|-----------|-------|
| *Architecture* | Codebook size | 400 |
| | Commitment cost ($\beta$) | 0.1 |
| | Decoder layers | 6 |
| | Decoder attention heads | 8 |
| | Feedforward dimension | 2048 |
| | Dropout | 0.1 |
| *Quantization* | Sampling method | Top-$k$ temperature sampling |
| | Top-$k$ | 5 |
| | Temperature ($\tau$) | 1.0 |
| | EMA decay ($\gamma$) | 0.99 |
| *Encoder* | $\alpha$ constraint | Adaptive, max 0.5 |
| *Training* | Optimizer | Adam |
| | Learning rate | 5e-3 |
| | Weight decay | 1e-4 (codebook and bias excluded) |
| | LR scheduler | ReduceLROnPlateau |
| | Batch size | 128 |
| | Max epochs | 100 (early stopping enabled) |
| | Random seed | 42 |

### A.3 Baseline Implementation Details

#### A.3.1 Clustering Baseline: LACOAT

We implement the Latent Concept Attribution (LACOAT) method from Yu et al. (2024), which discovers latent concepts through hierarchical clustering of contextualized representations.

**Concept Discovery.** For each word $w_i$ in the training dataset $\mathcal{D}$, we extract all contextualized representations $\vec{z}_{w_i}$ from layer $l$ using NeuroX (Dalvi et al., 2023). Following Yu et al. (2024), we filter words with frequency $< 5$ and randomly sample up to 20 contextual occurrences per word. Agglomerative hierarchical clustering is then applied using squared euclidean distance and ward's minimum-variance criterion to obtain $K = 400$ cluster centroids $\{\mathbf{c}_j\}_{j=1}^K$, each representing a latent concept.

**Concept Assignment.** At inference, a logistic regression classifier (ConceptMapper) maps salient token representations to their nearest cluster. The classifier is trained using cross-entropy loss with L2 regularization, the lbfgs solver, and 100 maximum iterations. We use Integrated Gradients with a zero-vector baseline and 500 approximation steps to identify salient tokens, selecting those that comprise 50% of the total attribution mass.

**Faithfulness Evaluation.** For concept ablation, we use the assigned cluster centroid $\mathbf{c}_j$ as the concept vector in our orthogonal projection framework (Appendix B.3). Note that while the original LACOAT implementation removes concepts through direct subtraction of the centroid vector, we employ orthogonal projection for more targeted concept removal.

### A.3.2 Cross-Layer Sparse Autoencoder

Following Dunefsky et al. (2024b), we implement a sparse autoencoder that learns to map representations from layer $l$ to layer $h$ through a sparse latent space.

**Architecture.** The encoder projects layer $l$ representations to a high-dimensional space, and the decoder reconstructs layer $h$:

$$\mathbf{h} = \mathrm{ReLU}(\mathbf{W}_{\mathrm{enc}}\mathbf{x} + \mathbf{b}_{\mathrm{enc}}) \tag{5}$$

$$\hat{\mathbf{y}} = \mathbf{W}_{\mathrm{dec}}\mathbf{h} + \mathbf{b}_{\mathrm{dec}} \tag{6}$$

where $\mathbf{W}_{\mathrm{enc}} \in \mathbb{R}^{d_{\mathrm{hidden}} \times d}$ with $d_{\mathrm{hidden}}$ set to $32 \times d$ for encoder models (24,576 for BERT/RoBERTa with $d = 768$) and $16 \times d$ for decoder models (65,536 for LLaMA with $d = 4096$; 32,768 for Qwen with $d = 2048$) as a practical memory compromise, following the expansion-ratio principle of Dunefsky et al. (2024b). We use untied weights ($\mathbf{W}_{\mathrm{enc}} \neq \mathbf{W}_{\mathrm{dec}}^T$) to reduce feature suppression (Bricken et al., 2023).

**Training.** The model minimizes the reconstruction loss with an $L_1$ penalty on activations:

$$\mathcal{L} = \|\mathbf{y} - \hat{\mathbf{y}}\|_2^2 + \lambda\|\mathbf{h}\|_1 \tag{7}$$

We set $\lambda = 1.4 \times 10^{-4}$ for encoder models (BERT/RoBERTa) and $\lambda = 1.5 \times 10^{-5}$ for decoder models (LLaMA/Qwen). We train using Adam (lr $= 5 \times 10^{-3}$, weight decay $10^{-4}$ for encoder weights and 0 for decoder weights), ReduceLROnPlateau scheduling (patience=5, factor=0.5), batch size 128, and early stopping (patience=15). The encoder bias is initialized to 0.1; the decoder bias and an input-centering bias $\mathbf{b}_{\mathrm{in}}$ are both initialized to the dataset mean of the respective representations.

**Concept Extraction.** For ablation, we identify the top-$k$ neurons by activation magnitude:

$$\mathcal{I}_k = \text{top-}k\{h_i\}_{i=1}^{d_{\mathrm{hidden}}} \tag{8}$$

We use the subspace spanned by their encoder weight vectors $\{\mathbf{e}_i\}_{i \in \mathcal{I}_k}$ (rows of $\mathbf{W}_{\mathrm{enc}}$) for orthogonal projection (Eq. 13). Encoder vectors are chosen because they live in the input (lower-layer) space of the token being perturbed (Dunefsky et al., 2024b; Bărbălau et al., 2025), unlike decoder vectors which map to the output (upper-layer) space. We report results for $k = 10$ in the main paper; Table 10 ablates over $k \in \{1, 5, 10\}$.

**Effect of Subspace Dimension $k$.** Table 10 shows SAE faithfulness results across $k \in \{1, 5, 10\}$. For encoder models, larger $k$ generally yields larger drops, confirming that SAE concepts are distributed across multiple neurons and benefit from multi-vector projection. For decoder models, performance remains near-zero regardless of $k$, suggesting that the limitation is architectural rather than the projection dimensionality.

### A.3.3 Single-Layer SAE.

The SAE counterpart of the Single-Layer VQ-VAE (Appendix A.3.4): identical to the cross-layer SAE above but reconstructing layer $l$ from itself, isolating the cross-layer contribution for the continuous baseline.

Table 10: SAE faithfulness (perturbed accuracy) across subspace dimension $k$. Lower is better (larger concept removal). Averaged over 3 seeds.

| Dataset | Model | $k = 1$ | $k = 5$ | $k = 10$ |
|---------|-------|---------|---------|----------|
| **ERASER-Movie** | RoBERTa | $0.5389 \pm 0.1052$ | $0.4085 \pm 0.0656$ | $0.4361 \pm 0.1664$ |
| | BERT | $0.6877 \pm 0.0378$ | $0.5024 \pm 0.0739$ | $0.4657 \pm 0.0733$ |
| | LLaMA | $0.9084 \pm 0.0012$ | $0.9092 \pm 0.0014$ | $0.9092 \pm 0.0007$ |
| | Qwen | $0.8828 \pm 0.0018$ | $0.8782 \pm 0.0037$ | $0.8731 \pm 0.0014$ |
| **Jigsaw** | RoBERTa | $0.9147 \pm 0.0026$ | $0.9164 \pm 0.0027$ | $0.9172 \pm 0.0022$ |
| | BERT | $0.8581 \pm 0.0167$ | $0.8432 \pm 0.0102$ | $0.8335 \pm 0.0064$ |
| | LLaMA | $0.8458 \pm 0.0021$ | $0.8444 \pm 0.0032$ | $0.8410 \pm 0.0012$ |
| | Qwen | $0.8475 \pm 0.0008$ | $0.8471 \pm 0.0022$ | $0.8463 \pm 0.0029$ |
| **AGNEWS** | RoBERTa | $0.3822 \pm 0.0244$ | $0.2900 \pm 0.0637$ | $0.3280 \pm 0.0807$ |
| | BERT | $0.3072 \pm 0.0140$ | $0.2875 \pm 0.0377$ | $0.3344 \pm 0.0103$ |
| | LLaMA | $0.9001 \pm 0.0009$ | $0.9001 \pm 0.0014$ | $0.8995 \pm 0.0010$ |
| | Qwen | $0.8994 \pm 0.0010$ | $0.8975 \pm 0.0030$ | $0.8992 \pm 0.0009$ |

#### A.3.4 Single-Layer VQ-VAE

This baseline uses identical architecture and hyperparameters as CLVQ-VAE but reconstructs layer $l$ from itself rather than mapping from layer $l$ to layer $h$, isolating the contribution of cross-layer analysis.

## B  Methodological Details

### B.1  EMA Update Details

During training, we perform temperature-based top-$k$ sampling to select codebook vectors, then apply EMA updates using hard assignments. For each codebook vector $\mathbf{e}_j$:

$$N_j^{(t)} = \gamma N_j^{(t-1)} + (1 - \gamma) \sum_i \mathbb{I}[j \text{ sampled for } \mathbf{z}_e^{(i)}] \tag{9}$$

$$\mathbf{m}_j^{(t)} = \gamma \mathbf{m}_j^{(t-1)} + (1 - \gamma) \sum_i \mathbf{z}_e^{(i)} \mathbb{I}[j \text{ sampled for } \mathbf{z}_e^{(i)}] \tag{10}$$

$$\mathbf{e}_j^{(t)} = \frac{\mathbf{m}_j^{(t)}}{N_j^{(t)}} \tag{11}$$

where $\gamma = 0.99$, $N_j^{(t)}$ tracks assignment counts, $\mathbf{m}_j^{(t)}$ accumulates vectors, and $\mathbb{I}[\cdot]$ indicates whether vector $j$ was selected via stochastic top-$k$ sampling. This provides stable codebook updates with improved utilization.

### B.2  Saliency Calculation Details

Our token saliency calculations are performed using the Layer Integrated Gradients (IG) method, which attributes a model's prediction back to its initial word embeddings. This approach allows us to see which input tokens were most important. However, because encoder and decoder-only models make predictions in fundamentally different ways, our attribution strategy is tailored to each architecture.

**Encoder-based Models (BERT and RoBERTa).**  For standard classification models like BERT and RoBERTa, the attribution process is straightforward. These models produce a final logit score for each class. We apply Layer IG to explain the logit of the predicted class, tracing its value back to the input embeddings. This directly measures how much each token contributed to the final classification decision.

**Decoder-only Models (LLaMA and Qwen).**  Decoder-only models are generative and perform next-token prediction. To adapt them for classification, we frame the task as having the model generate a single

token representing the class label (e.g., "0" or "1") immediately following the input prompt. The saliency calculation, therefore, aims to explain why the model generated that specific class token. We target the logit of the predicted class token and attribute its value back to the embeddings of the original prompt. This reveals which parts of the input text were most responsible for steering the model's generation towards the final class label.

### B.3 Orthogonal Projection Details

In our faithfulness evaluation, we remove concepts from sentence representations using orthogonal projection. Given a sentence representation $\mathbf{x} \in \mathbb{R}^d$ and a concept vector $\mathbf{z}_c \in \mathbb{R}^d$, we compute the perturbed representation as:

$$\mathbf{x}_{\text{perturbed}} = \mathbf{x} - \text{proj}_{\mathbf{z}_c}(\mathbf{x}) = \mathbf{x} - \frac{\mathbf{x} \cdot \mathbf{z}_c}{\|\mathbf{z}_c\|^2} \mathbf{z}_c \tag{12}$$

where $\text{proj}_{\mathbf{z}_c}(\mathbf{x})$ denotes the orthogonal projection of $\mathbf{x}$ onto $\mathbf{z}_c$, and $\mathbf{x} \cdot \mathbf{z}_c$ represents the dot product. The concept vector $\mathbf{z}_c$ corresponds to a codebook vector for VQ-VAE-based methods, a cluster centroid for the clustering baseline, or an encoder weight vector for SAE.

For the SAE baseline, we generalize this to a $k$-dimensional subspace. Given the top-$k$ encoder weight vectors $\{\mathbf{e}_{i_1}, \ldots, \mathbf{e}_{i_k}\}$ ranked by activation magnitude of their corresponding hidden units, we first orthonormalize them via Gram-Schmidt to obtain an orthonormal basis $\{\mathbf{u}_1, \ldots, \mathbf{u}_k\}$, then remove the projection onto their span:

$$\mathbf{x}_{\text{perturbed}} = \mathbf{x} - \sum_{j=1}^{k} (\mathbf{x} \cdot \mathbf{u}_j) \, \mathbf{u}_j \tag{13}$$

We use encoder weight vectors (rows of $\mathbf{W}_{\text{enc}}$) rather than decoder vectors because they live in the input-layer space of the CLS token being perturbed (Dunefsky et al., 2024b; Bărbălau et al., 2025), ensuring geometrically consistent ablation. This removes the component of $\mathbf{x}$ lying in the subspace spanned by the $k$ most active SAE features, providing a fairer perturbation that accounts for concept splitting across multiple neurons.

## C Faithfulness Evaluation

### C.1 Faithfulness Probe Implementation Details

The probe is a 2-layer MLP (Linear $\rightarrow$ ReLU $\rightarrow$ Dropout(0.2) $\rightarrow$ Linear) trained with Adam (lr = 0.001) and cross-entropy loss. We use 20-fold stratified cross-validation, preserving the class distribution in each fold. The best checkpoint per fold is selected by validation accuracy on the **Original CLS** embeddings, with early stopping (patience = 5, min. improvement $\delta = 0.001$, max. 100 epochs). The same checkpoint is then applied to the **Perturbed CLS** and **Random Perturbed CLS** variants of the held-out fold without any retraining.

In addition to the random direction baseline, we run a **Random Active Codebook** control (Appendix C.4), which removes a randomly sampled *active* codebook vector (one actually assigned in the dataset) rather than the predicted one. The random vector is sampled with probability proportional to its distance from the true assigned vector, and results are averaged over 5 independent samples. This verifies that accuracy drops under **Perturbed CLS** are specific to the concept-aligned direction rather than a consequence of removing any learned codebook direction.

### C.2 Reference Baseline Values

Table 11 provides the original CLS and random perturbed CLS accuracy values used as baselines across all faithfulness evaluation experiments. These values serve as reference points for calculating performance drops when salient concepts are removed from sentence representations.

Table 11: Reference baseline values for faithfulness evaluation across different model-dataset combinations and layer pairs.

| Model | Dataset | Layer Pair | Original CLS | Random Perturbed CLS |
|-------|---------|:----------:|:------------:|:--------------------:|
| RoBERTa | ERASER-Movie | 8–12 | 0.8777 | 0.8190 |
| RoBERTa | Jigsaw | 8–12 | 0.9121 | 0.9121 |
| RoBERTa | AGNews | 8–12 | 0.7275 | 0.6875 |
| BERT | ERASER-Movie | 8–12 | 0.8248 | 0.8237 |
| BERT | Jigsaw | 8–12 | 0.8995 | 0.8995 |
| BERT | AGNews | 8–12 | 0.7458 | 0.7433 |
| LLaMA-2-7b | ERASER-Movie | 28–32 | 0.9051 | 0.9039 |
| LLaMA-2-7b | Jigsaw | 28–32 | 0.8428 | 0.8407 |
| LLaMA-2-7b | AGNews | 28–32 | 0.8900 | 0.8900 |
| Qwen2.5-3B | ERASER-Movie | 32–36 | 0.8727 | 0.8751 |
| Qwen2.5-3B | Jigsaw | 32–36 | 0.8312 | 0.8363 |
| Qwen2.5-3B | AGNews | 32–36 | 0.8875 | 0.8883 |

## C.3   Llama Scope Sparse Transcoder

To verify whether the near-zero faithfulness drops for SAE on decoder models reflect a limitation of the Dunefsky implementation or a more general property of SAE-based methods, we additionally evaluate against a Llama Scope aligned sparse transcoder (He et al., 2024). Llama Scope uses TopK activation (hard sparsity, $k = 32$ active features per token) instead of ReLU+L1, and input/output normalization scaled to $\sqrt{d}$. We train this model on the same decoder-model activations (LLaMA-2-7B layers 28–32, Qwen2.5-3B layers 32–36) as the Dunefsky baseline and evaluate faithfulness using top-10 subspace projection.

Table 12 shows that Llama Scope produces similar trends to the Dunefsky baseline: near-zero or negative drops for LLaMA on ERASER-Movie and Jigsaw, and larger drops for Qwen on Jigsaw (16.4%). These results confirm that the weak decoder-model faithfulness is not an artifact of the Dunefsky implementation, but reflects a general limitation of SAE-based methods in capturing single-vector concept representations for decoder models — consistent with the feature-splitting hypothesis (Bricken et al., 2023) and motivating the discrete bottleneck approach of CLVQ-VAE.

Table 12: Llama Scope sparse transcoder faithfulness ($k = 10$ subspace projection, seed 42). Lower SAE perturbed accuracy indicates stronger concept removal.

| Dataset | Model | Original CLS | SAE Perturbed | Drop |
|---------|-------|:------------:|:-------------:|:----:|
| **ERASER-Movie** | LLaMA | 0.9119 | 0.9084 | 0.4% |
| | Qwen | 0.8797 | 0.8820 | -0.3% |
| **Jigsaw** | LLaMA | 0.8437 | 0.8562 | -1.5% |
| | Qwen | 0.8484 | 0.7089 | 16.4% |
| **AGNEWS** | LLaMA | 0.8984 | 0.8924 | 0.6% |
| | Qwen | 0.8975 | 0.8858 | 1.3% |

## C.4   Random Active Codebook Control

A potential concern with the faithfulness evaluation is that removing *any* learned codebook vector — not just the identified salient concept — would cause similar performance degradation, which would weaken the specificity interpretation. To test this, for each sentence we ablate 5 other active codebook vectors sampled using inverse-distance weighting (preferentially selecting vectors farther from the assigned concept direction) and report the average perturbed accuracy.

Table 13 shows that the identified salient concept causes substantially greater performance degradation than other active codebook vectors in 8 out of 9 configurations. In several cases, ablating a random active vector produces accuracy near the random perturbation baseline — other learned directions, while meaningful for other sentences, carry little relevance for the current one. This confirms that the drops in Table 1 reflect the targeted removal of a sentence-specific concept, not the incidental effect of perturbing any learned direction.

Table 13: Ablating the identified salient concept versus other active codebook vectors. Lower perturbed accuracy indicates greater importance. Random perturbed is an unstructured baseline.

| Model | Dataset | Salient Concept | Random Active Codebook | Random Perturbed |
|-------|---------|-----------------|------------------------|------------------|
| RoBERTa | ERASER-Movie | **0.0594** | 0.6197 | 0.8190 |
|  | Jigsaw | **0.6127** | 0.8886 | 0.9121 |
|  | AGNews | **0.0992** | 0.2863 | 0.6875 |
| BERT | ERASER-Movie | **0.5311** | 0.7764 | 0.8237 |
|  | Jigsaw | **0.7372** | 0.8606 | 0.8995 |
|  | AGNews | **0.6492** | 0.7365 | 0.7433 |
| Qwen | ERASER-Movie | **0.6113** | 0.7059 | 0.8751 |
|  | Jigsaw | **0.5809** | 0.6522 | 0.8363 |
|  | AGNews | 0.7536 | **0.7140** | 0.8883 |

# D Interpretability Evaluation

The following sections expand on the interpretability analyses summarized in the main paper. We provide complete metric definitions, the prompt template used for LLM judges, inter-judge agreement analysis across all configurations, and full codebook concept specificity results spanning all four models and three datasets.

## D.1 Evaluation Metric Formulations

The following metrics are used to evaluate concept quality in the LLM-as-a-Judge and human evaluation studies.

### D.1.1 LLM-as-a-Judge Metrics

1. **Mean Rating.** For each method $m$, the mean rating is computed as:

$$\text{MeanRating}(m) = \frac{1}{|C| \cdot |I| \cdot |J|} \sum_{c \in C} \sum_{i \in I} \sum_{j \in J} r_{m,c,i,j} \tag{14}$$

   where $C$ is the set of configurations (dataset-architecture pairs), $I$ is the set of instances, $J$ is the set of judges, and $r_{m,c,i,j}$ is the rating assigned by judge $j$ to method $m$ on instance $i$ in configuration $c$.

2. **Mean Reciprocal Rank (MRR).** For each configuration $c$, methods are ranked by their mean rating, with rank 1 assigned to the best-performing method. The MRR is:

$$\text{MRR}(m) = \frac{1}{|C|} \sum_{c \in C} \frac{1}{\text{rank}_{m,c}} \tag{15}$$

   where $\text{rank}_{m,c}$ is the rank of method $m$ in configuration $c$. Higher MRR indicates more consistent top performance across diverse settings.

3. **Win Rate.** The proportion of pairwise comparisons where a method achieves a higher mean rating than its competitor:

$$\text{WinRate}(m) = \frac{1}{|C| \cdot (|M| - 1)} \sum_{c \in C} \sum_{m' \neq m} \mathbb{1}[\text{MeanRating}_{m,c} > \text{MeanRating}_{m',c}] \times 100\% \tag{16}$$

where $C$ is the set of configurations, $M$ is the set of methods, $m' \neq m$ denotes all methods except $m$, and $\mathrm{MeanRating}_{m,c}$ is the average rating for method $m$ in configuration $c$.

4. **Kendall's W.** Kendall's coefficient of concordance measures agreement among judges on ranking methods. Given $n$ judges ranking $m$ methods:

$$W = \frac{12S}{n^2(m^3 - m)} \tag{17}$$

where $S = \sum_{i=1}^{m} \left( R_i - \frac{n(m+1)}{2} \right)^2$ and $R_i = \sum_{j=1}^{n} r_{ij}$ is the sum of ranks assigned to method $i$ across all $n$ judges. Values of $W \geq 0.7$ indicate strong inter-judge consensus.

### D.1.2 Human Evaluation Metrics

1. **Fleiss' Kappa ($\kappa$).** The measure of inter-annotator agreement beyond chance, ranging from -1 (worse than chance) to 1 (perfect agreement). We use the standard Fleiss' Kappa formula for $N$ instances, $n$ annotators, and $k$ categories.

2. **Average Confidence.** The mean confidence score across all instances and visualizations, where each annotator rated their confidence on a scale from 1 (lowest) to 10 (highest).

3. **Model Alignment Rate.** The proportion of cases where annotator predictions match the model's actual prediction:

$$\mathrm{Alignment}(m) = \frac{1}{|A| \cdot |I|} \sum_{a \in A} \sum_{i \in I} \mathbb{1}[\mathrm{pred}_{a,i}^{(m)} = \mathrm{pred}_{\mathrm{model},i}] \times 100\% \tag{18}$$

where $A$ is the set of annotators, $I$ is the set of instances, $\mathrm{pred}_{a,i}^{(m)}$ is annotator $a$'s sentiment prediction for instance $i$ based on method $m$'s word cloud visualization, and $\mathrm{pred}_{\mathrm{model},i}$ is the model's actual prediction for that instance.

## D.2 Human Evaluation: Sample Size and Design

Interpreting word-cloud visualizations of a model's internal concepts is cognitively demanding, so a larger pool risks annotator fatigue and reduced reliability. To control for selection bias, the 19 sentences were drawn through stratified random sampling with roughly equal representation of TP, TN, FP, and FN cases, so each prediction category is covered and the same sentence set is judged by all 14 annotators, allowing us to evaluate each instance from multiple perspectives. The Fleiss' Kappa of 0.864 provides strong statistical evidence of reliability at this scale. For precedent, our direct baseline LACOAT uses a small human evaluation equal to 50 sentences evaluated by 4 annotators. Finally, our LLM-as-judge evaluation (Section 4.2.2) addresses the small sample size concern directly: it extends concept quality judgments across all model-dataset configurations with an ensemble of four LLM judges across 100 samples, bridging the gap that a small human study cannot cover.

## D.3 LLM-as-a-Judge Prompt and Details

This section provides the exact prompt template used for the LLM-as-a-Judge evaluation.

### D.3.1 Prompt Template

The following template was provided to each LLM judge. Placeholders like `{sentence}` were populated dynamically for each sample.

```
You are an expert AI and Linguistics researcher. Your task is to evaluate how well
each "Concept Representation" explains a model's prediction for a given sentence.
```

**Context:**
- Sentence: "{sentence}"
- Model's Prediction: The model classified this as '{prediction}'
  (Meaning: {label_meaning}).

**Your Task:**
For each "Concept Representation" below, rate how well it provides a plausible reason
for the model's prediction. A concept representation is a group of words or sentences
that together represent a meaningful concept.

**Key Question:** If a model only focused on this "Concept Representation", how well
would it support making a prediction of '{label_meaning}'?

**Important Guidelines:**
- Similar representations with significant overlap should receive the same rating –
  if two concepts contain many of the same words or convey similar meanings, they
  should be rated equally.
- Words are not inherently better than sentences – concept sentences may be more
  detailed, but focus on the final sentiment/meaning inferred from the concept rather
  than the level of detail.
- Be flexible with pattern matching – as long as the overall concept or general theme
  can be identified and reasonably supports the prediction, it should be considered
  a good concept even if not perfectly precise.

{guidance_text}

**Rating Rubric:**
- 3 (Good): The concept representation shows a general connection to the predicted
  label '{label_meaning}' – even if not perfectly precise, the overall theme or
  pattern is recognizable and plausibly supportive.
- 2 (Fair): The concept representation has some connection to the prediction but may
  be broad, mixed, or only partially relevant.
- 1 (Poor): The concept representation shows little to no connection to the prediction,
  is mostly irrelevant, or clearly contradicts the expected label.

**Concept Representations to Evaluate:**
---
[This section is dynamically generated based on the configurations]

Concept from Configuration: "{config_1_name}"
- Concept Words: {words_for_config_1}

Concept from Configuration: "{config_2_name}"
- Representative Sentences:
  - "{sentence_1_for_config_2}"
  - "{sentence_2_for_config_2}"

[...additional configurations...]
---

**Output Instructions:**
Respond with a single valid JSON object. Use each configuration name as a key, with
the value being an object containing:
- 'rating': An integer from 1-3 based on the rubric above

- 'reason': A brief explanation justifying your rating

```
Example Format:
{
  "Config_A": {
    "rating": 3,
    "reason": "Contains words that strongly support the predicted label and form a
               coherent concept"
  },
  "Config_B": {
    "rating": 2,
    "reason": "Somewhat supports the prediction but contains mixed concepts"
  },
  "Config_C": {
    "rating": 1,
    "reason": "Does not support the prediction, contains irrelevant words"
  }
}
```

CRITICAL: You must respond with ONLY valid JSON in the exact format requested above.
Do not include any explanatory text before or after the JSON. Your entire response
should be parseable JSON.

### D.3.2 Dataset-Specific Guidance

The {guidance_text} placeholder in the prompt was populated with the following instructions depending on the dataset to provide task-specific context to the LLM judges.

**Jigsaw Toxicity Detection.**

```
**Guidance for Toxicity Detection:**
Be lenient with borderline cases since non-toxic sentences can be confused with toxic
ones. Context matters greatly - strong emotions, passionate language, or criticism
doesn't automatically mean toxicity. For 'Toxic' predictions: Look for patterns
suggesting harmful intent, but accept that detection is challenging. For 'Non-toxic'
predictions: Accept concepts suggesting civil discourse, even if emotionally charged
or critical. Sarcasm and irony can be easily misinterpreted.
```

**ERASER-Movie Sentiment Analysis.**

```
**Guidance for Sentiment Analysis:**
Movie reviews are often nuanced and mixed. For positive predictions: Accept concepts
suggesting overall appreciation, enjoyment, or recommendation, even if some criticisms
are present. For negative predictions: Accept concepts suggesting overall disappointment
or criticism, even if some positive aspects are mentioned. Focus on the dominant
sentiment direction rather than requiring pure positive/negative language.
```

**AGNews Topic Classification.**

```
**Guidance for Topic Classification:**
News topics frequently overlap - a tech company's earnings (Business + Science/Tech),
sports business deals (Sports + Business), or international conflicts affecting markets
(World + Business) are common. Accept concepts that show connection to the predicted
category even if they could reasonably fit multiple categories. Look for: World News
(countries, politics, conflicts, international themes), Sports (teams, games, players,
```

athletic activities), Business (companies, markets, financial concepts), Science/Tech (technology, research, innovations, technical concepts).

### D.4 Inter-Judge Agreement Analysis

To validate the reliability of our LLM-as-a-judge evaluation, we calculated Kendall's coefficient of concordance ($W$) across our ensemble of judges. Table 14 presents the agreement for the baseline comparison (corresponding to Section 4.2.1), and Table 15 presents the agreement for the initialization analysis (corresponding to Section 4.3.1).

Table 14: Inter-judge agreement on baseline rankings measured by Kendall's coefficient of concordance.

| Dataset | Kendall's W | Agreement Level |
|---|---|---|
| Jigsaw | 0.900 | Strong |
| ERASER-Movie | 0.475 | Weak |
| AGNews | 0.225 | Weak |
| **Overall Average** | **0.533** | **Moderate** |

Table 15: Inter-judge agreement on initialization method rankings measured by Kendall's coefficient of concordance.

| Dataset | Kendall's W | Agreement Level |
|---|---|---|
| Jigsaw | 0.910 | Strong |
| ERASER-Movie | 0.639 | Moderate |
| AGNews | 0.843 | Strong |
| **Overall Average** | **0.793** | **Strong** |

### D.5 Codebook Concept Specificity Analysis

This section provides the full methodology and complete results for the concept specificity analyses summarized in Section 4.2.1.

#### D.5.1 Methodology

**Label Purity (Vector-Level Analysis).** For each codebook vector $c$, let $T_c$ be the multiset of tokens assigned to it at inference time. Each token carries the label of its source sentence. Label purity is defined as

$$\text{purity}(c) = \frac{\max_l |T_c^{(l)}|}{|T_c|},$$

the fraction of tokens originating from the dominant label class. Vectors with fewer than five assigned tokens are excluded to avoid noise from near-empty entries. A random baseline is obtained by shuffling all token-to-vector assignments while preserving per-vector counts; for a balanced binary task this converges to 0.50 and for a balanced four-class task to 0.25. We additionally report the fraction of vectors reaching purity = 1.0 (single-class vectors) and compare mean purity against this baseline.

**Label-Conditioned Polysemy (Token-Level Analysis).** For each surface token $w$ appearing $\geq 10$ times across the evaluation set, we build a per-label code distribution $P_w^{(l)}$ over codebook indices. We apply a structured disambiguation criterion, requiring the top-3 most frequent codes to cover $\geq 50\%$ of the token's occurrences; this filters out tokens that spread uniformly across many vectors (indicating unstructured routing) and retains only those with interpretable, concentrated routing patterns. The mean pairwise JSD

across all label pairs is then

$$\overline{\mathrm{JSD}}(w) = \binom{|L|}{2}^{-1} \sum_{i<j} \mathrm{JSD}\left(P_w^{(l_i)} \| P_w^{(l_j)}\right),$$

where JSD (Lin, 1991) is a symmetric, bounded variant of KL divergence with base-2 logarithms giving range $[0, 1]$. Tokens with $\overline{\mathrm{JSD}} > 0$ and different dominant codes across at least one label pair are labeled *label-divergent*; the percentage of such tokens among all content tokens (*Label-Div. Tok.*) and their mean JSD are reported.

**Sentence Pair Discrimination (Sentence-Level Analysis).** For each sentence $s_i$, let $C_i$ be the set of unique codebook indices assigned to its tokens (stop words excluded). TF-IDF vectors (Sparck Jones, 1972) are computed over the evaluation set vocabulary with standard tokenisation. For all pairs $(i, j)$ with TF-IDF cosine similarity $\geq \tau$, we compute the Jaccard overlap $|C_i \cap C_j|/|C_i \cup C_j|$ and test whether same-label pairs have significantly higher overlap than different-label pairs using a one-sided Mann-Whitney $U$ test (Mann & Whitney, 1947); we report the ratio of median same-label to median different-label overlap (*SL/DL Ratio*). Conditioning on lexical similarity isolates semantic from surface effects: a positive ratio at high $\tau$ indicates that codebook overlap tracks meaning even when surface vocabulary is controlled. We evaluate at two thresholds $\tau \in \{0.1, 0.3\}$: the lower threshold captures more pairs (higher statistical power) while the higher threshold focuses on near-duplicate sentences where surface form is controlled most tightly.

**Qualitative Examples.** For the top-ranked label-divergent tokens by $\overline{\mathrm{JSD}}$, we identify the dominant codebook vector per label class and retrieve the most frequent co-occurring tokens assigned to each vector from the training set. This provides direct evidence that identical surface forms are routed to semantically distinct regions of the codebook depending on the surrounding context and prediction label. Table 16 shows representative examples at the maximum JSD = 1.0 (no overlap between per-label distributions) for all three datasets (RoBERTa).

### D.5.2 Full Results

Table 17 reports all metrics for all twelve model-dataset configurations. The pattern is consistent: every configuration shows label purity above its random baseline and positive same-label/different-label overlap ratios, all statistically significant at $p < 0.05$ or better (the majority at $p < 0.0001$). RoBERTa consistently shows the strongest ratios across all three datasets, which is expected given that it is fine-tuned on the target task and thus develops more label-discriminative intermediate representations. BERT, Qwen, and LLaMA also exhibit clear signal on all three analyses, though with smaller ratios, reflecting that the codebook specificity property holds across both encoder and decoder architectures. The three exceptions at the strictest threshold ($\tau = 0.3$) — Jigsaw/BERT ($p = 0.013$), Jigsaw/LLaMA ($p = 0.015$), and Jigsaw/Qwen ($p = 0.037$) — remain statistically significant at $p < 0.05$; the weaker signal at $\tau = 0.3$ is expected since fewer sentence pairs satisfy the stricter lexical similarity criterion, reducing statistical power.

Figures 2 and 3 show representative results for RoBERTa. In the purity histograms (Figure 2), the VQ-VAE mean (blue dashed) lies well above the random baseline (red dashed), and a visible spike at purity = 1.0 confirms the existence of single-class vectors. In the scatter plots (Figure 3), the trend lines for same-label and different-label pairs diverge as TF-IDF similarity increases, showing that the codebook overlap gap grows precisely when surface vocabulary is most similar — a pattern consistent with label-sensitive rather than surface-form routing.

## E   Design Choice Analysis

### E.1   Cross-Attention Ablation Study

We ablate the cross-attention mechanism to assess its impact on codebook learning and concept quality. Table 18 compares CLVQ-VAE with and without cross-attention across all models and datasets. We measure two metrics: (1) **Faithfulness**: perturbed CLS accuracy after concept removal (lower values indicate

Table 16: Top label-divergent tokens for ERASER-Movie, Jigsaw, and AGNews (RoBERTa). Each row shows one example sentence per label, the dominant codebook vector the token routes to, and the majority class among all tokens assigned to that vector (purity %).

| Dataset | Token | Example sentence | Vector | Majority (purity) |
|---|---|---|---|---|
| ERASER-Movie | *entertainment* | "the movie does not serve as a serious thriller nor as comic entertainment (because of its serious tone)." *(neg.)* | #137 | Neg. 91% |
| | | "Tarantino twists this age-old genre to produce over-the-top entertainment." *(pos.)* | #73 | Pos. 97% |
| | *simply* | "Matthew Modine is quite simply terrible." *(neg.)* | #49 | Neg. 93% |
| | | "Princess Caraboo is simply an eminently enjoyable entertainment." *(pos.)* | #246 | Pos. 94% |
| Jigsaw | *thank* | "We can take our time considering the wider issues. Thank you!" *(non-tox.)* | #265 | Non-tox. 100% |
| | | "Thank you for blocking him. That guy has been vandalizing the page for at least 2 weeks." *(toxic)* | #399 | Toxic 62% |
| | *far* | "As far as I know, the reverted edits by xxx are the default produced by the Undo tool." *(non-tox.)* | #54 | Non-tox. 100% |
| | | "Dear ClueBot NG, You suck. I am far more better." *(toxic)* | #170 | Toxic 100% |
| AGNews | *process* | "President Musharraf and PM Aziz met to review the composite dialogue process between India and Pakistan." *(World)* | #139 | World 57% |
| | | "Mike Williams is all but certain not to play Saturday due to delays in USC's appeal process." *(Sports)* | #23 | Sports 98% |
| | *sydney* | "Four years ago in Sydney, the US gymnasts had gone medal-free at the Olympics for the first time in 28 years." *(Sports)* | #204 | Sports 84% |
| | | "Sons of Gwalia (Sydney), the world's leading tantalum supplier, appointed outside managers after failing to reach agreement with creditors." *(Business)* | #304 | Business 80% |

concepts are more critical to model predictions), and (2) **Codebook Quality**: average pairwise cosine similarity between codebook vectors (lower values indicate more orthogonal, distinct concepts).

Including cross-attention yields lower cosine similarity in 8 out of 9 configurations and lower faithfulness scores in 7 out of 9 configurations. The improved codebook quality (more orthogonal vectors) and stronger faithfulness (larger performance drops upon ablation) indicate that cross-attention enables the discrete bottleneck to learn more distinct, task-critical concepts.

## E.2 Adaptive Alpha Parameter Study

Table 19 shows the impact of different $\alpha$ strategies on training dynamics and codebook utilization across training epochs on ERASER-Movie dataset and RoBERTa model.

Table 17: Full codebook concept specificity results across all twelve model-dataset configurations. **Label-Div. Tok.**: percentage of label-divergent content tokens. **Mean JSD**: computed over those tokens. **SL/DL Ratio**: same-label to different-label Jaccard overlap ratio at two TF-IDF cosine thresholds.

| Dataset | Model | Label Purity | Rand. | Label-Div. Tok. | Mean JSD | SL/DL Ratio $\tau = 0.1$ | $\tau = 0.3$ |
|---------|-------|-------------|-------|-----------------|----------|------------------|------------|
| ERASER-Movie | RoBERTa | 0.691 | | 75.8% | 0.848 | 4.26× | 8.93× |
| | BERT | 0.667 | 0.50 | 31.5% | 0.560 | 1.29× | 3.89× |
| | Qwen | 0.694 | | 44.0% | 0.690 | 1.09× | 2.11× |
| | LLaMA | 0.761 | | 37.9% | 0.628 | 1.05× | 1.43× |
| Jigsaw | RoBERTa | 0.646 | | 32.1% | 0.916 | 7.56× | 30.56× |
| | BERT | 0.601 | 0.50 | 33.5% | 0.560 | 1.55× | 1.68×* |
| | Qwen | 0.621 | | 33.2% | 0.666 | 1.17× | 1.18×* |
| | LLaMA | 0.659 | | 28.2% | 0.691 | 1.14× | 1.17×* |
| AGNews | RoBERTa | 0.353 | | 43.7% | 0.823 | 2.10× | 1.72× |
| | BERT | 0.357 | 0.25 | 54.2% | 0.623 | 1.28× | 1.33× |
| | Qwen | 0.389 | | 58.2% | 0.741 | 1.47× | 1.30× |
| | LLaMA | 0.465 | | 52.4% | 0.754 | 1.24× | 1.20× |

* $p < 0.05$ only at $\tau = 0.3$; all other entries $p < 0.001$.

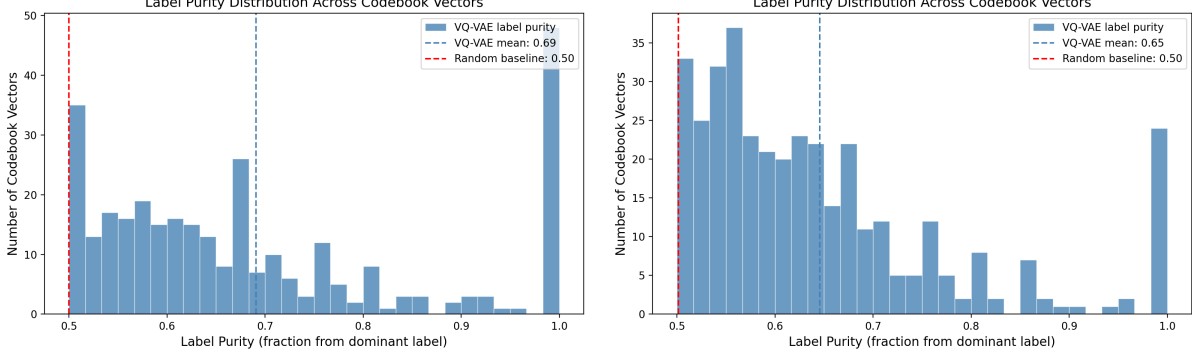

Figure 2: Label purity distributions for ERASER-Movie/RoBERTa (left) and Jigsaw/RoBERTa (right). The red dashed line marks the random baseline; the blue dashed line marks the VQ-VAE mean. The spike at purity = 1.0 represents vectors firing exclusively on one class.

These results reveal that training of adaptive $\alpha$ behaves like a curriculum mechanism: it begins with low values that preserve the original input embeddings and gradually increases to allow more expressive transformations. For instance, the limited adaptive setting starts around 0.28 and converges to 0.45, achieving high perplexity from early epochs and maintaining it consistently. This facilitates effective concept discovery while optimizing for low validation loss. In contrast, fixed low $\alpha$ values such as 0.1 retain high perplexity but restrict the model's ability to adapt representations, resulting in higher loss. On the other hand, fixed high $\alpha$ values (e.g., 0.75 or 1.0) take significantly longer to reach useful perplexity levels, delaying convergence. Notably, when $\alpha = 0$, the encoder remains an identity function throughout training and becomes decoupled from decoder and quantization gradients, leading to stagnation.

We limit the adaptive $\alpha$ to do a maximum of 0.5 change. We noticed that allowing complete change of input embedding using adaptive $\alpha$ resulted in a high $\alpha$ which reduced final perplexity.

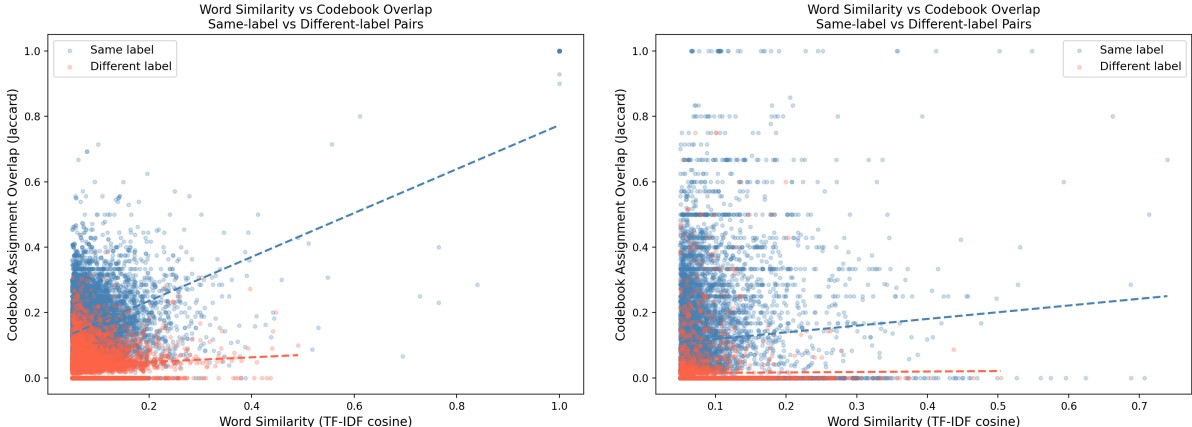

Figure 3: Word similarity (TF-IDF cosine) vs. codebook assignment overlap (Jaccard) for same-label (blue) and different-label (red) sentence pairs, for ERASER-Movie/RoBERTa (left) and Jigsaw/RoBERTa (right). Dashed trend lines are fitted separately per group. The growing divergence between same- and different-label trend lines confirms that the codebook discriminates by meaning above and beyond surface vocabulary.

Table 18: Impact of cross-attention on faithfulness and codebook quality. With Res includes cross-attention; No Res removes it. Lower values indicate better performance for both metrics. Bold indicates better performance.

| Model | Dataset | Faithfulness (With vs. No Res) | Cosine Sim. (With vs. No Res) |
|---|---|---|---|
| RoBERTa | ERASER-Movie | 0.0594 vs 0.0560 | **0.751** vs 0.924 |
| RoBERTa | Jigsaw | 0.6127 vs **0.5152** | 0.575 vs **0.484** |
| RoBERTa | AGNews | **0.0992** vs 0.1067 | **0.906** vs 0.976 |
| BERT | ERASER-Movie | **0.5311** vs 0.7560 | **0.479** vs 0.760 |
| BERT | Jigsaw | **0.7372** vs 0.8752 | **0.312** vs 0.666 |
| BERT | AGNews | **0.6492** vs 0.7442 | **0.597** vs 0.839 |
| Qwen | ERASER-Movie | **0.6113** vs 0.7254 | **0.684** vs 0.690 |
| Qwen | Jigsaw | **0.5809** vs 0.5934 | **0.719** vs 0.746 |
| Qwen | AGNews | **0.7536** vs 0.8033 | **0.687** vs 0.790 |

### E.3 Commitment Weight Analysis

Table 20 shows the impact of commitment cost $\beta$ on codebook utilization across ERASER and Jigsaw datasets for RoBERTa model.

Table 20: Impact of commitment cost $\beta$ on validation perplexity across datasets.

| Commitment Cost ($\beta$) | ERASER Perplexity | Jigsaw Perplexity |
|---|---|---|
| 0.0 | 213.45 | 163.45 |
| 0.1 | 210.26 | 164.07 |
| 0.3 | 189.74 | 145.65 |
| 0.6 | 170.76 | 81.38 |
| 1.0 | 23.94 | 30.71 |

Table 19: Alpha parameter analysis showing perplexity evolution across training epochs and final validation loss.

| Alpha Strategy | Initial | Epoch 10 | Epoch 30 | Final | Best Val Loss |
|---|---|---|---|---|---|
| Adaptive (Limited) | 1.97 | 198.5 | 216 | 210.6 | 0.033 |
| Adaptive (Complete) | 1.86 | 25.54 | 50.70 | 63.21 | 0.033 |
| Fixed $\alpha$=0.0 | 238.1 | 237.3 | 239.8 | 237.2 | 0.045 |
| Fixed $\alpha$=0.1 | 180.9 | 232.3 | 238.1 | 230.8 | 0.040 |
| Fixed $\alpha$=0.4 | 1.95 | 126.9 | 160.2 | 157.2 | 0.036 |
| Fixed $\alpha$=0.75 | 1.077 | 1.883 | 27.517 | 40.106 | 0.033 |
| Fixed $\alpha$=1.0 | 1.155 | 2.397 | 15.219 | 147.470 | 0.032 |

Higher $\beta$ values force stronger commitment to assigned codebook vectors, reducing perplexity but limiting concept diversity. Lower $\beta$ values allow more flexible assignments, promoting diverse concept identification. $\beta$=0.1 achieves optimal balance between concept diversity and training stability.

### E.4 Sampling Parameter Analysis

We analyze the impact of temperature and top-k parameters on codebook utilization and concept identification performance. Table 21 shows validation perplexity across different configurations, while Table 22 presents faithfulness evaluation results.

Table 21: Impact of temperature and top-k parameters on codebook utilization (validation perplexity) for ERASER-Movie on RoBERTa model. Higher perplexity indicates more diverse codebook usage.

| Temperature | Top-k | Validation Perplexity |
|---|---|---|
| 0.5 | 5 | 207.14 |
| 1.0 | 5 | 210.63 |
| 2.0 | 5 | 217.14 |
| 3.0 | 5 | 220.09 |
| 1.0 | 1 | 207.07 |
| 1.0 | 10 | 210.37 |
| 1.0 | 50 | 211.57 |
| 1.0 | 100 | 212.51 |

Increasing temperature from 0.5 to 3.0 increases validation perplexity from 207.14 to 220.09, reflecting greater exploration in codebook selection. For top-k values with $\tau = 1.0$, perplexity increases more gradually from 207.07 (k=1) to 212.51 (k=100), indicating that the exploration-exploitation balance shifts more gradually compared to temperature adjustments.

Table 22: Impact of temperature and top-k values on concept identification performance for ERASER-Movie on RoBERTa model. Despite significant differences in sampling parameters, perturbed CLS accuracies remain within a narrow range (0.0783–0.0911).

| Top-k | Temperature | Perturbed CLS Accuracy |
|:-----:|:-----------:|:----------------------:|
| 1 | 1.0 | 0.0911 |
| 10 | 1.0 | 0.0864 |
| 100 | 1.0 | 0.0817 |
| 400 | 1.0 | 0.0877 |
| 400 | 0.1 | 0.0806 |
| 400 | 1.0 | 0.0877 |
| 400 | 2.0 | 0.0783 |
| 400 | 4.0 | 0.0911 |

*Reference values:*
Original CLS: 0.7604    Random Perturbed CLS: 0.7264

While temperature and top-k parameters significantly affect codebook utilization (as measured by perplexity), perturbed accuracy exhibits limited sensitivity to these hyperparameters. They varied only within a narrow 0.0783–0.0911 range across all configurations, despite substantial differences in codebook usage patterns. This suggests that while different sampling strategies lead to different concept distributions, the resulting concepts remain comparably important for model predictions. We adopt conservative values ($\tau = 1.0$, k=5) to balance stable training dynamics with reasonable codebook diversity.

### E.5   Layer Pair Analysis

We analyze the impact of layer pair selection on concept discovery by evaluating several combinations across RoBERTa and BERT models fine-tuned on ERASER-Movie, Jigsaw, and AGNews. Our final choice of layers 8–12 is motivated both by theoretical understanding and empirical evidence.

**Theoretical Motivation.** Transformer-based architectures such as RoBERTa and BERT are known to exhibit hierarchical processing, where lower layers capture surface-level linguistic patterns and intermediate layers encode rich semantic information (Yu et al., 2024). We hypothesize that the transformation between Layers 8 and 12 best captures the transition from semantically meaningful representations to task-specific decision-making features. While our method is layer-agnostic in design, selecting this range enables optimal interpretability.

**Empirical Validation.** We conducted systematic experiments across multiple layer pairs, comparing the impact of concept removal using our perturbation-based faithfulness metric. Table 23 presents accuracy after perturbing the [CLS] token using discovered concepts, alongside baselines using original and randomly perturbed inputs.

Table 23: Layer pair analysis showing that layers 8–12 capture the most meaningful transformations across all datasets.

| Model-Dataset | Layer Pair | Perturbed CLS | Original CLS | Random Perturbed |
|---|---|---|---|---|
| RoBERTa–ERASER | 0–4 | 0.5140 | 0.4988 | 0.5012 |
| RoBERTa–ERASER | 4–8 | 0.7069 | 0.5374 | 0.5269 |
| RoBERTa–ERASER | 8–12 | 0.0583 | 0.8777 | 0.8190 |
| RoBERTa–Jigsaw | 0–4 | 0.4962 | 0.4962 | 0.4962 |
| RoBERTa–Jigsaw | 4–8 | 0.1734 | 0.7692 | 0.7653 |
| RoBERTa–Jigsaw | 8–12 | 0.5853 | 0.9121 | 0.9121 |
| RoBERTa–AGNews | 0–4 | 0.2567 | 0.2500 | 0.2500 |
| RoBERTa–AGNews | 4–8 | 0.3575 | 0.4092 | 0.3783 |
| RoBERTa–AGNews | 8–12 | 0.0967 | 0.7275 | 0.6875 |
| BERT–ERASER | 0–4 | 0.5035 | 0.5012 | 0.5024 |
| BERT–ERASER | 4–8 | 0.7593 | 0.7642 | 0.7631 |
| BERT–ERASER | 8–12 | 0.4813 | 0.8248 | 0.8237 |
| BERT–Jigsaw | 0–4 | 0.4962 | 0.4936 | 0.4936 |
| BERT–Jigsaw | 4–8 | 0.8154 | 0.8919 | 0.8906 |
| BERT–Jigsaw | 8–12 | 0.7308 | 0.8995 | 0.8995 |
| BERT–AGNews | 0–4 | 0.2408 | 0.2500 | 0.2500 |
| BERT–AGNews | 4–8 | 0.7283 | 0.8125 | 0.8175 |
| BERT–AGNews | 8–12 | 0.7117 | 0.7458 | 0.7433 |

These results support three key observations. First, early layers (0–4) encode minimal task-relevant concepts, as perturbing them leads to no meaningful change in prediction accuracy. Second, the 4–8 layer range shows inconsistent behavior across datasets–on ERASER-Movie, we observe an unexpected accuracy increase following concept perturbation, possibly due to the removal of noisy or redundant features, whereas on Jigsaw and AGNews, the expected performance drop suggests some level of concept relevance. Finally, the 8–12 configuration consistently reveals meaningful concepts across all datasets: perturbing this range significantly degrades model performance, indicating that it captures the most faithful and impactful transformations from semantic features to final task-specific representations.

### E.6 Computational Complexity Analysis

We analyze the computational complexity of CLVQ-VAE compared to SAE during training. Let $B$ denote batch size, $L$ sequence length, $D$ model dimension, $H_{sae}$ the SAE hidden dimension, $K$ the codebook size, and $N$ the number of transformer decoder layers.

### E.6.1 SAE Complexity

The SAE performs encoder projection, ReLU activation, and decoder projection:

$$\mathcal{O}_{\text{SAE}} = \mathcal{O}(B \cdot L \cdot D \cdot H_{sae}) \tag{19}$$

where $H_{sae}$ ranges from $16 \cdot D$ to $32 \cdot D$ depending on model size (24,576 for $D = 768$; 65,536 for $D = 4096$) and must be substantially larger than $D$ for feature disentanglement. Complexity scales linearly with sequence length but is dominated by large matrix multiplications.

### E.6.2 CLVQ-VAE Complexity

CLVQ-VAE consists of encoder transformation, vector quantization, and transformer decoder:

$$\mathcal{O}_{\text{enc+quant}} = \mathcal{O}(B \cdot L \cdot D^2) + \mathcal{O}(B \cdot L \cdot D \cdot K) \tag{20}$$

$$\mathcal{O}_{\text{decoder}} = \mathcal{O}(N \cdot B \cdot (L^2 \cdot D + L \cdot D^2)) \tag{21}$$

$$\mathcal{O}_{\text{total}} = \mathcal{O}(B \cdot L \cdot D \cdot K) + \mathcal{O}(N \cdot B \cdot (L^2 \cdot D + L \cdot D^2)) \tag{22}$$

Complexity scales quadratically with sequence length due to self-attention in the $N = 6$ decoder layers. Vector quantization remains efficient with compact codebook size $K = 400$.

## F   Qualitative Analysis

To show how CLVQ-VAE represents concepts related to sentiment analysis, we examine examples from the ERASER-Movie review dataset. We present word clouds generated by our method for different prediction outcomes.

### F.1   False Negative Example

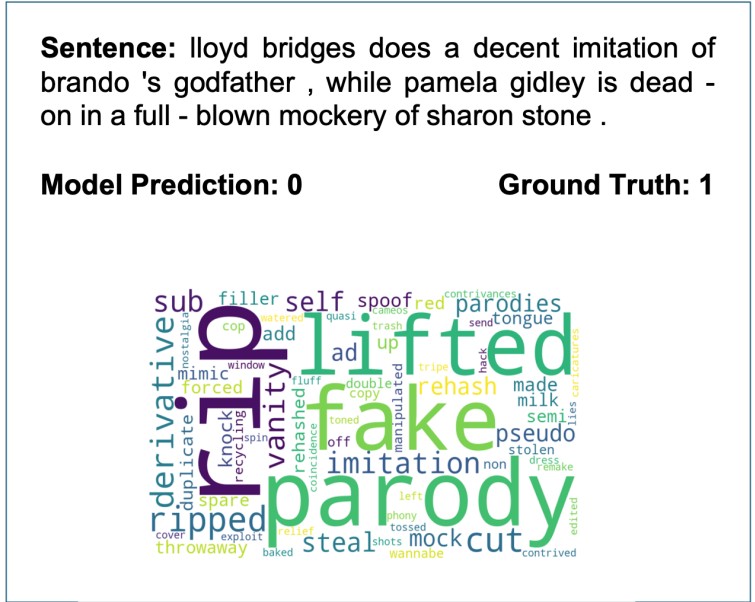

Figure 4: False negative example (Model: 0, Ground Truth: 1) showing concept clusters related to imitation and parody.

In Figure 4, we show a false negative example where the model incorrectly predicts negative sentiment for a positive review. The review describes actors performing imitations, with Lloyd Bridges doing a "decent imitation of Brando's godfather" and Pamela Gidley performing a "dead-on mockery of Sharon Stone".

The word cloud in Figure 4 contains many terms related to imitation ("lifted", "fake", "parody", "imitation", "ripped"). Despite the review framing these imitations positively as "decent" and "dead-on", the model associates these imitation concepts with negative sentiment. This shows a limitation in distinguishing between criticism of unoriginality and praise for good impersonations.

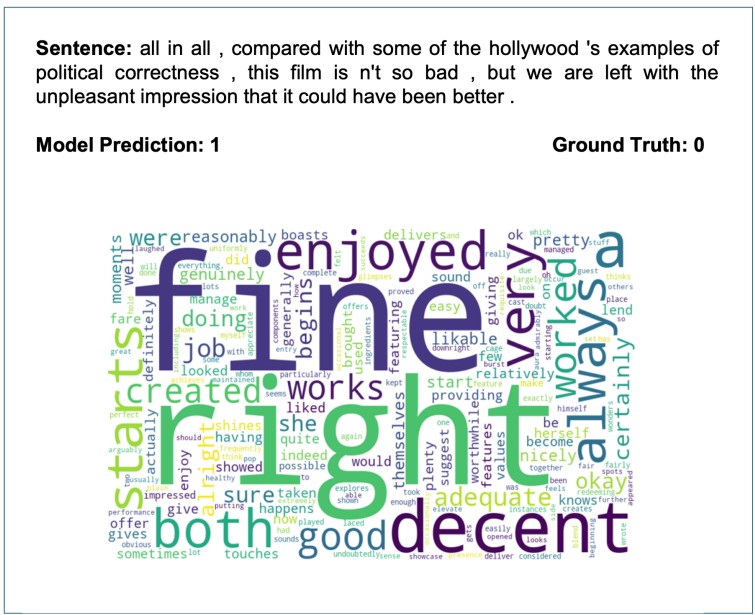

Figure 5: False positive example (Model: 1, Ground Truth: 0) showing terms of moderate approval despite the overall negative sentiment.

### F.2 False Positive Example

Figure 5 shows a false positive case where the model incorrectly predicts positive sentiment for a negative review. The review describes a film that "isn't so bad" but leaves "the unpleasant impression that it could have been better".

The word cloud in Figure 5 shows terms of moderate approval ("fine", "enjoyed", "decent", "right", "good"). The model focused on the mild praise while missing the more subtle negative sentiment. This shows a limitation in distinguishing between faint praise and genuine positive sentiment in reviews with mixed language.

### F.3 True Negative Example

Figure 6 shows a true negative example where the model correctly predicts negative sentiment. The review states "this film has neither the quality of cinematography nor the moments of glory to be highlighted".

The word cloud in Figure 6 contains terms expressing absence or lack ("nothing", "barely", "whatsoever", "little", "nowhere", "zero"). This shows how CLVQ-VAE effectively captures concepts related to deficiency, correctly identifying negative sentiment in the review.

### F.4 True Positive Example

In Figure 7, we show a true positive example where the model correctly predicts positive sentiment for "the acting is superb from everyone involved". This direct praise is an ideal case for sentiment analysis.

The word cloud in Figure 7 shows many positive descriptors ("outstanding", "fantastic", "awesome", "magnificent"). This demonstrates how our model captures related positive terms, particularly for straightforward expressions of praise.

These examples show how CLVQ-VAE captures discrete concepts for sentiment classification, which often includes sentiment-laden terms and their semantic relationships. The false prediction cases (Figures 4 and 5) highlight limitations identified by CLVQ-VAE in RoBERTa model.

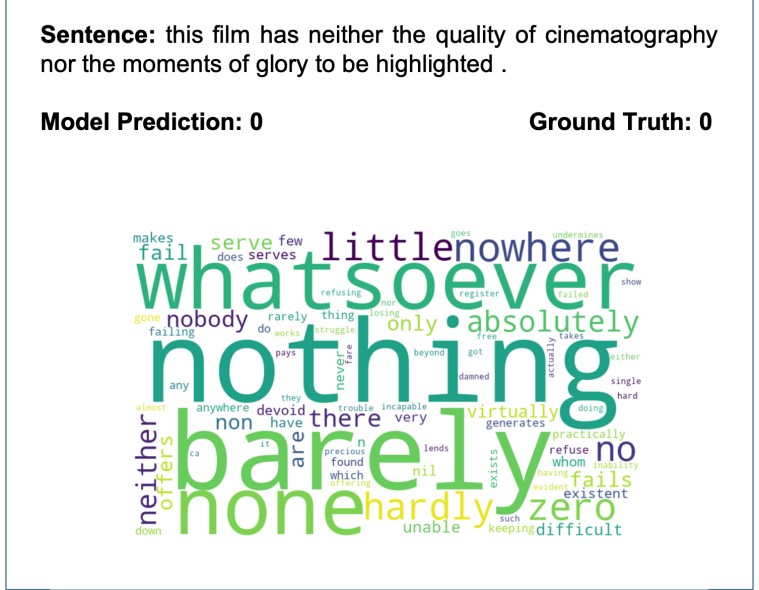

Figure 6: True negative example (Model: 0, Ground Truth: 0) showing terms expressing absence or deficiency.

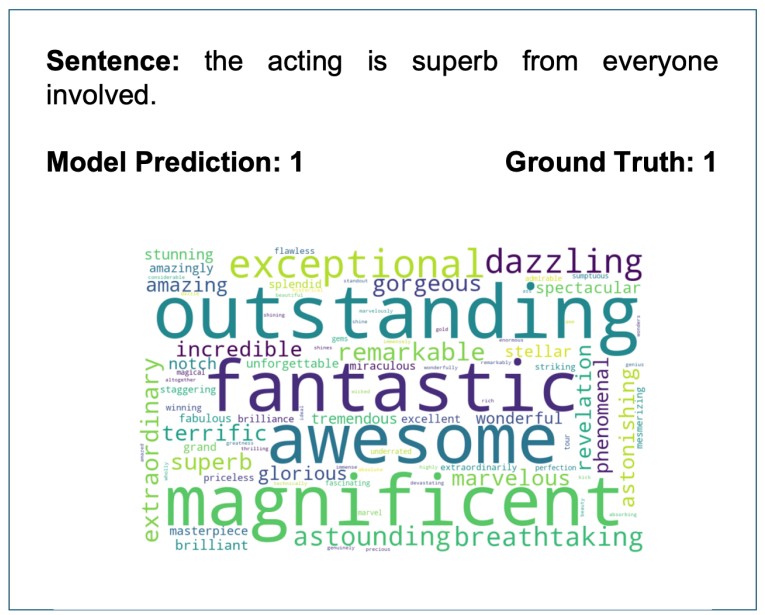

Figure 7: True positive example (Model: 1, Ground Truth: 1) showing positive descriptors for "the acting is superb from everyone involved".

