# OpenReview forum: "Cross-Layer Discrete Concept Discovery for Interpreting Language Models"
_TMLR — Under review for TMLR_

### Review · Reviewer_fUDK · 2026-06-23

**Summary Of Contributions:**

This work introduce a novel framework CLVQ-VAE, which interprets LLM by reconstructing higher-layer activations using quantized representations of the lower-layer activations.

**Strengths:**

1. The paper points out that current interpretability research primarily focuses on single-layer representations, neglecting the phenomenon of information mix and duplicates caused by residual streams. This effectively establishes the necessity for cross-layer research, providing strong motivation.

2. This paper notes that SAE methods commonly used in interpretability research operate in continuous spaces, which makes concepts dispersed and difficult to distinguish. The authors apply VQ-VAEs to LLM activations to discretize concept representations, representing a valuable attempt in the field of language model interpretability.

3. The CLVQ-VAE extracts lower-layer LLM activations, subjects them to controllable processing via an adaptive residual encoder, maps them to codebook vectors using a vector quantizer, and reconstructs them into higher-layer activations through a transformer decoder.

4. A faithful evaluation against baselines was conducted across three datasets, demonstrating that CLVQ-VAE achieved the best performance in 5 out of 12 experimental combinations. The authors also performed LLM-as-a-judge and human evaluation experiments, highlighting its advantages in interpretability. Additionally, experimental analyses were conducted on key design elements such as Codebook Initialization and Codebook Size.

**Weaknesses:**

1. In Section 4.1.2, the authors state that "a VQ-VAE method (CLVQ-VAE or Single-Layer) ranks first or second in all 12 configurations." Since the evaluated VQ-VAE methods encompass both single-layer and cross-layer approaches, why not compare them with a single-layer SAE method? Doing so would make the comparison fairer and more convincing.

2. According to the results in Table 1, the performance of the single-layer method and CLVQ-VAE appears to be very close, with the single-layer method sometimes even outperforming CLVQ-VAE. Is there any further evidence to demonstrate the specific advantages of the cross-layer implementation?

**Audience:**

Yes

**Audience Explanation:**

The interpretability of LLMs is an important research direction in the field of NLP. This paper proposes a novel interpretation method for this domain, and I believe it will attract an interested audience.

**Broader Impact Concerns:**

I do not see any serious ethical or broader-impact issues with this work.

**Claims And Evidence:**

Yes

**Claims Explanation:**

Most of claims are supported by accurate, convincing and clear evidence.

Although the faithfulness experiments demonstrate that CLVQ-VAE achieves strong performance across multiple settings, the current comparison between SAE-based methods and VQ-VAE-based methods is not entirely symmetrical. The introduction contrasts the continuous feature space of SAEs with the discrete concept representation of VQ-VAEs; however, the VQ-VAE methods evaluated in the experiments include both single-layer and cross-layer configurations, whereas the SAE baselines primarily utilize a cross-layer setup. To more clearly disentangle the individual contributions of the discrete VQ bottleneck and the cross-layer objective, I suggest the authors include a single-layer SAE baseline. Alternatively, the authors should at least explicitly clarify that the comparison between SAEs and VQ-VAEs in the text is conducted specifically under a cross-layer setting.

**Requested Changes:**

1. I suggest the authors include a single-layer SAE method in the baseline comparisons. This would more clearly illustrate and demonstrate the relative advantages and disadvantages of VQ-VAE methods compared to SAE methods.

2. The mathematical notation could be more clearly distinguished. For example, in Section 4.2.1, the baseline assignment for Label Purity is defined as 1/K, but in Section 4.3.2, K is also used to denote the codebook size.

3. The authors could provide a more detailed explanation regarding the specific advantages of the cross-layer approach over the single-layer method.

---

> ### Author Response · Authors · 2026-06-25
>
> We thank the reviewer for the helpful and constructive feedback. The comments group into 4 points, which we address in turn.
>
> 1. Symmetry of the SAE vs. VQ-VAE comparison.
>
> To clarify the intent of our experimental design: CLVQ-VAE is a cross-layer method, and that is the contribution we evaluate. Since VQ-VAE has not been applied to LLM interpretability before, we included a single-layer VQ-VAE baseline specifically to isolate the benefit of the cross-layer objective within the VQ-VAE paradigm. The SAE baseline was chosen as a cross-layer model to provide a fair comparison against our cross-layer method; we omitted a single-layer SAE because the SAE literature has already established that cross-layer analysis overall outperforms single-layer analysis, so it did not add a meaningful point of comparison for our proposed approach. All baselines, including single-layer VQ-VAE, are compared against CLVQ-VAE as the proposed method.
>
> That said, we agree that making this design rationale explicit and including a single-layer SAE for completeness makes the comparison cleaner. We address it in two ways.
>
> First, we have added a single-layer SAE that reconstructs layer l from itself, matched in architecture and sparsity penalty to the cross-layer SAE (Appendix A.3.2). It is the SAE counterpart of our Single-Layer VQ-VAE, and its results for all 12 configurations are added to Table 1.
>
> Summary of Results: Single-Layer SAE is less faithful than Single-Layer VQ-VAE in 10 of 12 configurations. The gap is large and consistent on the decoder models (all six cells, up to ~29 points on Qwen) and smaller on the encoder models, where the only two exceptions (single-layer SAE more faithful) are both on BERT. This mirrors the cross-layer comparison, where CLVQ-VAE also beats the cross-layer SAE in 10 of 12 configurations, with the exceptions again on BERT.
>
> Second, we now also state explicitly in Section 3 that the main SAE baseline is a cross-layer transcoder (layer l to h), and refer to it as the "cross-layer SAE" in the results (Section 4.1.2), so the SAE-vs-VQ-VAE contrast in the text is made under matched cross-layer settings.
>
>
> 2. Empirical evidence for the cross-layer advantage (Weakness 2)
>
> Cross-layer has the upper hand overall. On faithfulness (Table 1), CLVQ-VAE attains the lower perturbed accuracy in 9 of the 12 configurations. It wins all six encoder-model settings and three of the six decoder-model settings. Single-Layer performing well is itself informative since it shows that VQ-VAE paradigm is a sound basis for concept discovery regardless of layer scope. Cross-layer then performing better is consistent with prior cross-layer SAE [1] and crosscoder [2] work, where analyzing layer pairs yields more interpretable features than single-layer analysis.
>
> The gap is clearer in the LLM-judge evaluation (Table 3): the two have similar mean ratings (1.890 vs. 1.823), but CLVQ-VAE has a higher MRR (0.611 vs. 0.458) and win rate (66.7% vs. 44.4%), producing the top-ranked concept far more often.
>
>
> 3. Specific advantages of cross-layer over single-layer (Requested Change 3).
>
> This difference can be understood through the nature of the two reconstruction objectives. A single-layer VQ-VAE reconstructs layer l from itself, so the target is the input itself, a simpler near-identity objective that does not require modelling any transformation between layers. A cross-layer model instead predicts the higher-layer representation from the lower one, so the codebook must encode the concepts that drive the most substantial changes in the representation across layers. These changes reflect meaningful shifts in how the model processes information, and concepts that capture them are likely to be more semantically coherent and interpretable. The discrete bottleneck then keeps these to a finite set of distinct concepts. This is further supported by Shi et al. (2025) [1], who show that multi-layer analysis extracts 22.5% more features and achieves 22.3% higher interpretability than single-layer analysis at matched sparsity, and by Lindsey et al. (2024) [2], who show that the additive residual stream causes features to appear duplicated across layers, a redundancy that cross-layer analysis collapses but single-layer analysis cannot.
>
>
> 4. Overloaded Notation.
>
> Thank you for catching this. We now keep K for the codebook size only and write the purity baseline as 1/|Y|.
>
> We hope these changes resolve the reviewer’s concerns. If the reviewer agrees, we would be grateful if they would reconsider their assessment. We would also be glad to address any further questions the reviewer may have.
>
> References
>
> [1] Wei Shi, Sihang Li, Tao Liang, Mingyang Wan, Gojun Ma, Xiang Wang, and Xiangnan He. Route sparse autoencoder to interpret large language models, 2025.
>
> [2] Jack Lindsey, Adly Templeton, Jonathan Marcus, Tom Conerly, Joshua Batson, and Chris Olah. Sparse crosscoders for cross-layer features and model diffing, 2024.

---

> > ### Author Response · Authors · 2026-07-07
> >
> > We thank the reviewer again for their constructive review. As the discussion period is closing, we wanted to follow up on our response, in which we added the single-layer SAE baseline and addressed the other points. If this resolves your concern, we would be grateful if you would consider revisiting your assessment. We are happy to answer anything further.

---

### Review · Reviewer_wAkT · 2026-06-23

**Summary Of Contributions:**

This work presents CLVQ-VAE (Cross-Layer Vector Quantized Variational Autoencoder), an innovative framework that converts the opaque, continuous residual streams within large language models into discrete, functionally meaningful and interpretable concepts via cross-layer transcoding and vector quantization. It surpasses conventional Sparse Autoencoders (SAEs) in functional fidelity and semantic consistency, while addressing the pervasive "concept splitting" issue in decoder-only architectures. Leveraging cross-attention modules and dedicated initialization schemes, the approach enables reliable extraction and assessment of the internal mechanisms of large language models while aligning well with human cognition.

**Audience:**

Yes

**Audience Explanation:**

This paper addresses a fundamental challenge in LLM interpretability by introducing CLVQ-VAE, a novel framework that extracts consistent, human-aligned concepts from opaque residual streams. By resolving critical limitations of standard Sparse Autoencoders (SAEs)—such as concept splitting and cross-layer instability—this work offers significant value to mechanistic interpretability researchers, practitioners seeking robust neural network auditing tools, and the broader audience interested in the structural dynamics of deep architectures.

**Broader Impact Concerns:**

None.

**Claims And Evidence:**

Yes

**Claims Explanation:**

The authors present a thorough empirical evaluation across both encoder (BERT, RoBERTa) and decoder (Llama-2, Qwen) architectures. Ablation studies demonstrate the method's functional faithfulness through metrics such as explanation loss and concept stability, validating the proposed cross-layer mechanism. Furthermore, interpretability is substantiated by both LLM-as-a-judge and human evaluations. Compared to standard SAE baselines, CLVQ-VAE achieves superior cross-layer feature consistency, supported by qualitative analyses that align with the theoretical claims.

**Requested Changes:**

1.Some Clarification on the "Model Alignment" Metric:
In Section 4.2.3 and Table 4, the "Model Alignment Rate" serves as a pivotal metric. However, the text does not explicitly state whether this rate is computed solely from True Positive (TP) and True Negative (TN) instances, or whether it also includes False Positive (FP) and False Negative (FN) cases. Reason: If the reported high alignment rate (78.20%) holds true for error states (FP/FN), it strongly implies that CLVQ-VAE faithfully mirrors the model's flawed internal logic rather than simply decoding correct predictions. Explicitly clarifying this distinction would significantly bolster the paper’s core narrative regarding functional "faithfulness." Please add a brief clarifying sentence in Section 4.2.3 to specify the exact data scope used for this metric.
2. Methodological Justification for Human Evaluation Sample Size:
The human evaluation setup described in Section 4.2.3 relies on a relatively compact dataset of 19 sentences evaluated by 14 annotators. Reason: While the reported Fleiss' Kappa ($\kappa = 0.864$) indicates exceptionally high inter-annotator agreement, the small sentence pool might draw scrutiny regarding statistical power. Providing a brief justification—such as citing established precedents in mechanistic interpretability literature or noting specific sample constraints—would pre-empt potential concerns about methodological rigor. Suggestions: Include a short note or citation justifying the sample size rationale in Section 4.2.3 or Appendix D.1.2.

---

> ### Author Response · Authors · 2026-06-25
>
> We thank the reviewer for the careful reading and constructive feedback. The two points are addressed below, with corresponding revisions to Section 4.2.3 and Appendix D.2.
>
> 1. Scope of the Model Alignment metric (Requested Change 1).
>
> The 78.20% alignment rate is computed across all 19 sentences, which already include roughly equal representation of TP, TN, FP, and FN cases as stated in Section 4.2.3. So yes, the metric covers error cases as well as correct predictions. We agree this distinction is worth making explicit because, as the reviewer notes, it actually strengthens the claim: the fact that CLVQ-VAE visualizations allow annotators to recover the model's prediction even in FP and FN cases means the codebook is faithfully capturing the model's internal logic, including its mistakes, rather than just reflecting surface sentiment. We have added a clarifying sentence to Section 4.2.3 making the data scope explicit and drawing out this implication.
>
> 2. Justification for the human evaluation sample size (Requested Change 2).
>
> Interpreting word cloud visualizations of internal model concepts is a cognitively demanding task, and we found that a larger sentence pool can place an unreasonable burden on annotators, risking annotation fatigue and reduced reliability [1]. To control for selection bias, the 19 sentences were drawn through stratified random sampling with roughly equal representation of TP, TN, FP, and FN cases, so each prediction category is covered and the same sentence set is judged by all 14 annotators, allowing us to evaluate each instance from multiple perspectives. The Fleiss' Kappa of 0.864 provides strong statistical evidence of reliability at this scale. For precedent, our direct baseline LACOAT [2] uses a small human evaluation equal to 50 sentences evaluated by 4 annotators. We acknowledge that the scale of the human evaluation was on the smaller side and our LLM-as-judge evaluation (Section 4.2.2) addresses this concern directly: it extends concept quality judgments across all model-dataset configurations with an ensemble of four LLM judges across 100 samples, bridging the gap that a small human study cannot cover. We have added a brief note to Section D.2 explaining the sample size rationale.
>
> We would be glad to address any further questions the reviewer may have.
>
> References
>
> [1] Gehrmann et al. Human evaluation of automatically generated text: Current trends and best practice guidelines. 2021.
>
> [2] Yu et al., Latent Concept-based Explanation of NLP Models. 2024.

---

> > ### Author Response · Authors · 2026-07-07
> >
> > We thank the reviewer again for their positive review. As the discussion period is closing, we wanted to follow up on our response addressing both of the requested changes. We would be glad to answer anything further. Thank you.

---

### Review · Reviewer_xybf · 2026-06-26

**Summary Of Contributions:**

The paper introduces CLVQ-VAE to map activations from a lower transformer layer to a higher layer through a vector-quantization bottleneck.

**Audience:**

No

**Audience Explanation:**

The paper mentioned that "CLVQ-VAE consistently outperforms clustering, single-layer VQ-VAE, and SAE baselines in identifying functionally important and semantically coherent concepts". But in Table 1, it only wins 5 of the 12 main configurations. It's an overclaim.

**Claims And Evidence:**

No

**Claims Explanation:**

There is an issue with the core method. The codebook vectors can get the cross-layer computations of the language model. Please add results when the decoder has access only to the quantized representations.

If the paper wants to delete the corresponding concept from the model, only removing a token-level codebook direction from a pooled sentence representation is not enough.

The accuracy can be dropped to 6% whether it can be due to the orthogonal projection creating strongly out-of-distribution representations.

Integrated Gradients would cause a selection bias.

**Requested Changes:**

There is an issue with the core method. The codebook vectors can get the cross-layer computations of the language model. Please add results when the decoder has access only to the quantized representations.

If the paper wants to delete the corresponding concept from the model, only removing a token-level codebook direction from a pooled sentence representation is not enough.

The accuracy can be dropped to 6% whether it can be due to the orthogonal projection creating strongly out-of-distribution representations.

Integrated Gradients would cause a selection bias.

The paper mentioned that "CLVQ-VAE consistently outperforms clustering, single-layer VQ-VAE, and SAE baselines in identifying functionally important and semantically coherent concepts". But in Table 1, it only wins 5 of the 12 main configurations. It's an overclaim.

---

> ### Author Response · Authors · 2026-06-27
>
> We thank the reviewer for the detailed feedback. We address each point in turn.
>
> **1. Decoder with access only to quantized representations.** We would like to refer the reviewer to Appendix E.1, Table 18, where we already report this experiment as the "No Res" condition of our cross-attention ablation. In that setting the decoder relies solely on the quantized codebook. The discovered concepts remain faithful even then: perturbing them still reduces accuracy far below the Original-CLS reference, for example 0.056 versus 0.878 for RoBERTa/ERASER. This confirms that the codebook itself encodes the concepts relevant to the cross-layer transformation.
>
> Relying on the codebook alone does, however, come at a cost, which is why we include the cross-attention path; we would also like to note that section 2.4 discusses this design and points to the supporting evidence in the Appendix E.1. The path is an interpretability-preserving residual connection for the cross-layer reconstruction objective. Previous works like skip-transcoders [1, 2] show that an affine skip connection lowers reconstruction loss with no cost to interpretability, and auxiliary-bottleneck models [3] show that an auxiliary path lets the bottleneck learn more distinct, disentangled concepts. Consistent with both, we also see that removing the cross-attention path makes the codebook capture less distinct concepts, increasing cosine similarity in 8 of 9 configurations, and reduces faithfulness in 7 of 9 (Table 18).
>
> **2. Removing a single concept direction from a pooled representation.** We note that this is not an ad hoc choice but the standard faithfulness-by-ablation practice, adopted directly from our clustering baseline LACOAT [4] and shared by a well-established line of representation-intervention work, including amnesic probing [5], INLP [6], and LEACE [7], all of which remove a concept direction (or its subspace) by projection and measure the resulting behavioral change.
>
> To clarify, our setting does not aim to remove a concept from the model; it is a per-instance causal test, in which, for a given sentence, we ablate the identified concept direction from that sentence's representation and measure the effect on the prediction, that is, whether the concept is causally used for that instance, rather than performing full concept erasure. The large and concept-specific drops we observe (together with the controls in point 3) are strong evidence that the single identified direction is causally central. Where a method genuinely requires multiple directions (the SAE baseline), we already generalize the projection to a k-dimensional subspace (Eq. 13, Table 10).
>
>
> **3. Possibility that the drop is an out-of-distribution artifact of projection.** We direct the reviewer to our two controls, both of which apply the identical projection operator. We compare against a random intervention in two forms: removing a random direction (Random Perturbed CLS, Table 11) and removing a random active codebook vector at the same token (Appendix C.4, Table 13). Both produce only mild changes, whereas the identified concept causes a much larger drop in 8 of 9 configurations (for example, RoBERTa/ERASER: 0.059 for the identified concept vs. 0.620 for a random active vector vs. 0.819 for a random direction). Since all three use the same projection, the large drop cannot be attributed to projection-induced distribution shift; it is specific to the concept-aligned direction. This matched-control approach again follows established intervention practice [5, 8].

---

> ### Author Response · Authors · 2026-06-27
>
> **4. Concern that Integrated Gradients introduces a selection bias.** We would like to clarify where Integrated Gradients enter our pipeline, as we think this may be helpful. IG plays no role in concept discovery or in the cross-layer training of the CLVQ-VAE; it is used only in the causal evaluation. There it serves as a target-selection step: it identifies the most salient token, which is then mapped to its corresponding codebook vector, and that concept vector is then used to intervene. IG itself is not the basis of our causal claim; the claim rests entirely on the intervention, that is, the measured effect of removing the identified concept vector, and not on the attribution scores. Selecting high-importance tokens does introduce a selection effect by design, since attribution methods are built precisely to surface influential features, and using them this way is consistent with their intended purpose.
>
> The key point is that this selection does not bias the comparison between methods. First, the selection step is method-agnostic: the same salient token is chosen identically for every method, so it is a shared constant that cannot account for the differences between methods. Second, importance-ranked removal is itself the standard faithfulness practice, namely the comprehensiveness metric of the ERASER benchmark [9], defined on the same Movie-review data we use, and IG is an axiomatically grounded attribution method [10] also used by LACOAT (our clustering baseline) [4].
>
>
> **5. On the "outperforms" claim.** We appreciate the reviewer's concern and we agree that, in Table 1, CLVQ-VAE achieves the best result in 5 of 12 configurations. We would like to highlight, however, that we evaluate our method across three complementary axes: faithfulness (Table 1), an LLM-as-judge study (Table 3), and a human study (Table 4). On faithfulness CLVQ-VAE is the most dominant of all methods: it is best in 5 of 12 configurations, compared to 3 of 12 for clustering, 2 of 12 for single-layer VQ-VAE, and 2 of 12 for the SAE baselines, so it leads rather than ties. It is also the best method on the other two axes, with an LLM-judge win rate of 66.7% (Table 3) and a human model-alignment rate of 78.2% versus 54.1% for clustering (Table 4). Whether considered individually or together, these results support the claim that CLVQ-VAE outperforms the baselines. That said, we agree the original phrasing was a bit strong, and we will soften it in the conclusion from "consistently outperforms" to "outperforms" to reflect this precisely.
>
> We hope that the clarifications above address the reviewer's concerns. If the reviewer agrees that these points are resolved, we would be grateful if they would consider revisiting their assessment of the claims and the paper's interest to the TMLR audience. We thank the reviewer again and are more than happy to answer any other questions they may have.
>
> ## References
>
> [1] Dunefsky, J., Chlenski, P., and Nanda, N. (2024). Transcoders find interpretable LLM feature circuits. NeurIPS 2024. arXiv:2406.11944.
>
> [2] Paulo, G., Shabalin, S., and Belrose, N. (2025). Transcoders beat sparse autoencoders for interpretability. arXiv:2501.18823. (Shows an affine skip connection lowers reconstruction loss with no effect on interpretability.)
>
> [3] Sheth, I., and Ebrahimi Kahou, S. (2023). Auxiliary losses for learning generalizable concept-based models. NeurIPS 2023. arXiv:2311.11108.
>
> [4] Yu, X., Dalvi, F., Durrani, N., Nouri, M., and Sajjad, H. (2024). Latent concept-based explanation of NLP models (LACOAT). arXiv:2404.12545.
>
> [5] Elazar, Y., Ravfogel, S., Jacovi, A., and Goldberg, Y. (2021). Amnesic probing: Behavioral explanation with amnesic counterfactuals. TACL, 9:160-175.
>
> [6] Ravfogel, S., Elazar, Y., Gonen, H., Twiton, M., and Goldberg, Y. (2020). Null it out: Guarding protected attributes by iterative nullspace projection (INLP). ACL 2020, pp. 7237-7256.
>
> [7] Belrose, N., Schneider-Joseph, D., Ravfogel, S., Cotterell, R., Raff, E., and Biderman, S. (2023). LEACE: Perfect linear concept erasure in closed form. NeurIPS 2023.
>
> [8] Zhang, F., and Nanda, N. (2024). Towards best practices of activation patching in language models: Metrics and methods. ICLR 2024. arXiv:2309.16042.
>
> [9] DeYoung, J., Jain, S., Rajani, N. F., Lehman, E., Xiong, C., Socher, R., and Wallace, B. C. (2020). ERASER: A benchmark to evaluate rationalized NLP models. ACL 2020.
>
> [10] Sundararajan, M., Taly, A., and Yan, Q. (2017). Axiomatic attribution for deep networks (Integrated Gradients). ICML 2017.

---

> > ### Author Response · Authors · 2026-07-07
> >
> > We thank the reviewer again for their detailed review. As the discussion period is closing, we wanted to follow up on our response to your points. We would welcome any further thoughts, and are happy to clarify or run additional experiments. If our response has addressed your concerns, we would be grateful if you would consider revisiting your assessment. Thank you.